# A shorter splicing isoform antagonizes ZBP1 to modulate cell death and inflammatory responses

Masahiro Nagata[1,2,4], Yasmin Carvalho Schäfer [1,2,4], Laurens Wachsmuth [1,2,4] &
Manolis Pasparakis [1,2,3✉]

## Abstract

**Z-DNA-binding protein 1 (ZBP1) is an interferon-inducible sensor of Z-DNA and Z-RNA, which has emerged as a critical regulator of cell death and inflammation. ZBP1 binds Z-DNA and Z-RNA via its Zα domains, and signals by engaging RIPK3 and RIPK1 via its RIP homotypic interaction motifs (RHIMs). Here, we show that mice express an alternatively-spliced shorter ZBP1 isoform (ZBP1-S), which harbours the Zα domains but lacks the RHIMs, and acts as an endogenous inhibitor of the full-length protein (ZBP1-L). Mice and cells expressing only ZBP1-S are resistant to ZBP1-mediated cell death and inflammation. In contrast, cells lacking ZBP1-S show increased ZBP1-L-induced death compared to cells expressing both isoforms. Moreover, loss of the short isoform accelerates and exacerbates skin inflammation induced by ZBP1-mediated necroptosis of RIPK1-deficient keratinocytes, revealing an important physiological role of ZBP1-S. Mechanistically, ZBP1-S suppresses ZBP1-L-mediated cell death by binding to Z-nucleic acids via its Zα domains. Therefore, ZBP1-S acts as an endogenous inhibitor that competes with full-length ZBP1-L for binding Z-nucleic acid ligands to fine-tune ZBP1-mediated cell death and inflammation.**

**Keywords** ZBP1; Necroptosis; Cell Death; Inflammation
**Subject Categories** Autophagy & Cell Death; Immunology; RNA Biology

## Introduction

Z-DNA binding protein 1 (ZBP1, also known as DAI or DLM-1) has emerged as a critical regulator of cell death, inflammation, and immunity (Maelfait and Rehwinkel, 2023). ZBP1 contains two Zα domains that bind double-stranded DNA and RNA with an alternative left-handed helical conformation termed the Z-form (Athanasiadis, 2012; Schwartz et al, 2001). Z-DNA and Z-RNA were originally discovered in structural studies of synthetic oligonucleotides more than forty years ago (Hall et al, 1984; Wang et al, 1979). However, the conditions under which Z-RNA and Z-DNA may form in cells as well as their biological function, have remained enigmatic (Athanasiadis, 2012; Herbert, 2019; Rich and Zhang, 2003). Moreover, ZBP1 contains two well-characterized and a third putative receptor-interacting protein (RIP) homotypic interaction motif (RHIM), which enable it to interact with other RHIM-containing proteins (Kaiser et al, 2008; Rebsamen et al, 2009). In addition to ZBP1, three more proteins containing RHIMs are expressed in mammalian cells, namely RIPK1, RIPK3, and TIR-domain-containing adapter-inducing interferon-β (TRIF) (Kaiser et al, 2008; Pasparakis and Vandenabeele, 2015; Rebsamen et al, 2009). ZBP1 engages RIPK3 and RIPK1 in a RHIM-dependent manner to trigger mixed lineage kinase-like (MLKL)-dependent necroptosis and caspase-8-dependent apoptosis. ZBP1 plays an important role in viral infections by inducing necroptosis and apoptosis that act to restrict viral replication but may also cause tissue damage and result in pathology (Koehler et al, 2017; Kuriakose et al, 2016; Thapa et al, 2016; Upton et al, 2012; Zhang et al, 2020). Importantly, sensing of viral Z-nucleic acids was shown to be required for activating ZBP1-mediated cell death and its function in antiviral defense both in vitro and in vivo in mouse models (Guo et al, 2018; Koehler et al, 2021; Maelfait et al, 2017; Sridharan et al, 2017; Zhang et al, 2020).

Studies in genetically engineered mouse models revealed an important role of ZBP1 in inducing cell death and inflammation in the absence of viral infection. ZBP1 was shown to cause keratinocyte necroptosis and severe skin inflammation in mice with epidermis-specific knockout of RIPK1 (RIPK1[E-KO]) (Dannappel et al, 2014; Lin et al, 2016). Furthermore, knock-in mice expressing RIPK1 with mutated RHIM (*Ripk1[mR/mR]*) died perinatally due to ZBP1-mediated RIPK3-MLKL-dependent necroptosis (Lin et al, 2016; Newton et al, 2016). Consistent with the genetic in vivo data, deficiency in RIPK1 or mutation of its RHIM-sensitized cells to ZBP1-mediated cell death. These studies revealed an important RHIM-dependent function of RIPK1 that is essential to counteract the activation of ZBP1 and the induction of necroptosis and inflammation in the skin during development. ZBP1 was also shown to cooperate with TNFR1 to induce intestinal epithelial cell (IEC) necroptosis and inflammation in mice with IEC-specific knockout of Fas-associated with death domain (FADD), an adapter that is essential for activation of caspase-8 downstream of death receptors (Schwarzer et al, 2020). In addition, inhibition of caspase-8 sensitized cells to ZBP1-mediated necroptosis

[1]Institute for Genetics, University of Cologne, D-50674 Cologne, Germany. [2]Cologne Excellence Cluster on Cellular Stress Responses in Aging-Associated Diseases (CECAD), University of Cologne, D-50931 Cologne, Germany. [3]Center for Molecular Medicine (CMMC), University of Cologne, D-50931 Cologne, Germany. [4]These authors contributed equally: Masahiro Nagata, Yasmin Carvalho Schäfer, Laurens Wachsmuth. ✉E-mail: pasparakis@uni-koeln.de

(Jiao et al, 2020; Rodriguez et al, 2022; Schwarzer et al, 2020; Yang et al, 2020), further supporting that FADD-caspase-8-dependent mechanisms inhibit ZBP1-induced necroptosis. Importantly, mutation or deletion of the ZBP1 Zα domains prevented skin inflammation and perinatal lethality in RIPK1[E-KO] and *Ripk1*[mR/mR] mice, respectively, as well as intestinal inflammation in FADD[IEC-KO] mice (Devos et al, 2020; Jiao et al, 2020; Kesavardhana et al, 2020). Furthermore, we showed previously that treatment of cells with nuclear export inhibitors sensitizes cells to ZBP1-mediated cell death (Jiao et al, 2020), which was subsequently confirmed in an independent study (Karki et al, 2021). Notably, deletion or mutation of both Zα domains or mutation of Zα2 alone prevented ZBP1-mediated cell death induced by RIPK1 deficiency or RHIM mutation, treatment with caspase-8 inhibitors, or nuclear export inhibitors (Jiao et al, 2020). Together, these findings demonstrated that ZBP1 is activated by sensing endogenous Z-nucleic acids (Devos et al, 2020; Jiao et al, 2020), most likely Z-RNA derived from endogenous retroelements (Jiao et al, 2020), via its Zα domains and induces necroptosis and inflammation. ZBP1 was also reported to cause intestinal epithelial cell death and inflammation in mice with IEC-specific knockout of the histone methyltransferase SETDB1 (Wang et al, 2020) and to mediate heatstroke-induced cell death and tissue pathology (Yuan et al, 2022), as well as anti-tumor responses (Cho et al, 2024; Karki et al, 2021; Zhang et al, 2022). Further studies suggested that ZBP1 is activated by sensing telomeric-repeat-containing RNA transcripts to cause replicative crisis (Nassour et al, 2023), and by mitochondrial DNA to induce cardiotoxicity (Lei et al, 2023). Moreover, three independent studies reported that ZBP1 causes early lethality and severe type I interferonopathy in mice with hemizygous ADAR1 Zα domain mutations (de Reuver et al, 2022; Hubbard et al, 2022; Jiao et al, 2022). Collectively, these studies revealed important functions of ZBP1 in inflammation and disease.

The expression of two different isoforms of ZBP1 in mouse cells was previously observed in immunoblot assays from different cell types, including mouse embryonic fibroblasts (MEFs), lung fibroblasts (LFs), and bone marrow-derived macrophages (BMDMs) (Jiao et al, 2022; Kesavardhana et al, 2020). However, the role of these isoforms and their functional interactions have remained unclear. Here we functionally characterize the two ZBP1 isoforms expressed in mouse cells: the full-length ZBP1 protein (ZBP1-Long, ZBP1-L) and the alternatively-spliced truncated ZBP1-short isoform (ZBP1-S) that only contains the two N-terminal Zα domains and lacks the C-terminal part containing the RHIMs. We show that ZBP1-S suppresses ZBP1-L-mediated cell death in a manner that depends on Zα domain binding to Z-nucleic acids. Importantly, we provide experimental evidence that endogenously produced ZBP1-S limits ZBP1-L-mediated cell death and inflammation in vivo in a mouse model of ZBP1-mediated skin inflammation. Collectively, these results revealed that the alternatively-spliced ZBP1-S isoform acts as a molecular decoy that binds Z-nucleic acids to fine-tune ZBP1 activation and the induction of downstream signaling, causing cell death and inflammation.

# Results

## Two ZBP1 isoforms are produced by alternative splicing

Two main murine ZBP1 transcripts are annotated in genomic databases: a transcript containing all 8 exons (ENSMUST00000029018.14) that is predicted to produce the full-length ZBP1 protein comprising 411 amino acids (ZBP1-L) (Uniprot ID Q9QY24-1), and a shorter transcript generated by utilization of an alternatively-spliced exon 4 within the third intron of the gene (ENSMUST00000109116.3) (Fig. 1A). This alternatively-spliced transcript is predicted to produce a truncated ZBP1 isoform (ZBP1-S) (Uniprot ID Q9QY24-2) comprising 187 amino acids including the two N-terminal Zα domains but lacking all C-terminal sequences containing the RHIMs (Fig. 1A). Analysis of RNA sequencing data from different mouse cell types including T cells, B cells, myeloid cells, adipocytes and fibroblasts (Tabula Muris et al, 2018) confirmed the expression of the two isoforms (Fig. 1B). RNA-seq read pile up alignment visualization revealed increased number of reads of alternative exon 4, which is unique to ZBP1-S, compared to exons 4–8 that are unique to ZBP1-L, suggesting higher mRNA expression of the short isoform compared to the long one in B cells, myeloid cells, adipocytes, and fibroblasts (Fig. 1B). The expression of ZBP1 peptides corresponding to the long and short isoforms was previously observed in immunoblot assays from different cell types including MEFs, LFs and BMDMs (Jiao et al, 2022; Kesavardhana et al, 2020). To comprehensively assess the expression of the two isoforms at the protein level, we performed immunoblot analysis of whole cell protein extracts from primary LFs and BMDMs as well as from thymus and spleen tissues isolated from wild type and *Zbp1*[−/−] mice (Figs. 1C–E and EV1A). Thymus and spleen tissues from wild-type mice expressed the long and short ZBP1 isoforms at similar levels (Fig. 1C). Primary LFs from wild-type mice showed weak basal expression of ZBP1-L, whereas we did not detect expression of either isoform in unstimulated BMDMs (Fig. 1D,E). However, after stimulation with IFNα for 24 h, both LFs and BMDMs showed robust expression of both isoforms, with ZBP1-S detected at slightly reduced levels compared to ZBP1-L (Fig. 1D,E). Consistently, mRNA expression analysis with RT-qPCR showed that both isoforms were strongly induced by IFNα stimulation for 20 h in primary LFs, with the short isoform showing a slightly lower fold change compared to the long one (Fig. EV1B). The mRNA of both isoforms degraded at a similar rate after IFNα removal (Fig. EV1B). Cycloheximide (CHX) chase experiments indicated that ZBP1-S may have a shorter half-life, although the difference to the long was rather marginal (Fig. EV1C). Moreover, proteasome inhibition mildly increased the protein levels of ZBP1-S, suggesting that the short isoform may undergo increased proteasomal degradation (Fig. EV1D).

## ZBP1-S inhibits ZBP1-L-induced cell death

Research on ZBP1 has primarily focused on the full-length isoform, which mediates downstream signaling via its RHIM domains, while the role of ZBP1-S has remained elusive. To investigate the physiological function of the two isoforms, we generated two different knock-in mouse models that express exclusively either ZBP1-L or ZBP1-S from the endogenous *Zbp1* genomic locus. To generate mice expressing exclusively ZBP1-L (*Zbp1*[L/L]), we used CRISPR/Cas9-mediated gene targeting to delete the intronic region encompassing the alternative exon 4, which is required for the production of ZBP1-S but is dispensable for the expression of ZBP1-L (Fig. 2A). To generate mice expressing exclusively ZBP1-S (*Zbp1*[S/S]), we used CRISPR/Cas9-mediated gene targeting to delete the genomic region encompassing exons 4–8 that encode the C-terminal part of the protein including the three RHIMs, resulting in a genomic locus that can only produce the short isoform (Fig. 2B). DNA sequencing confirmed the correct targeting of the endogenous *Zbp1* genomic locus in both *Zbp1*[L/L] and *Zbp1*[S/S] mice

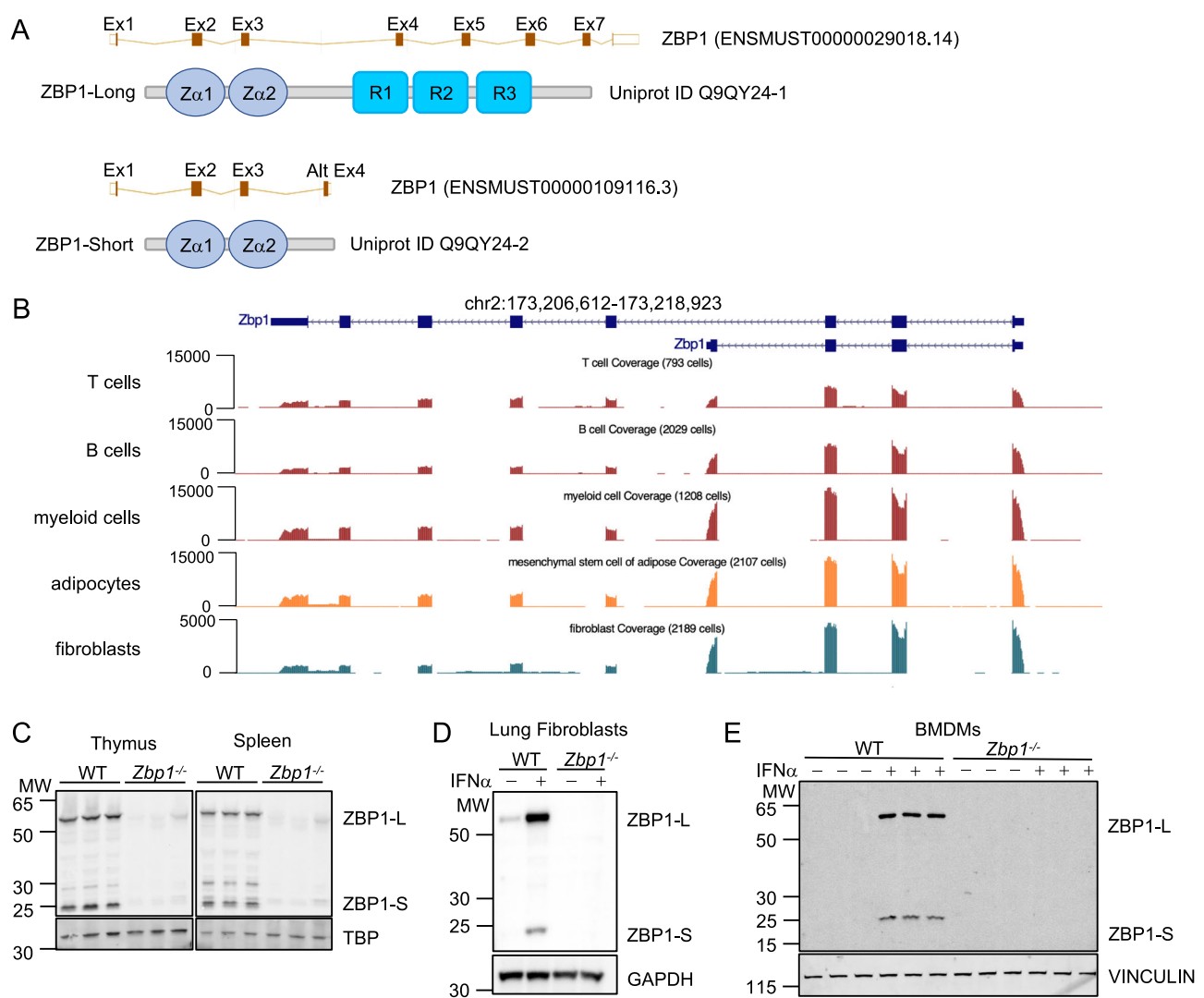

**Figure 1. Mice express an alternatively-spliced short ZBP1 isoform in addition to the full-length protein.**

(A) Schematic depicting the alternatively-spliced transcript variants of murine *Zbp1*, which encode the full-length protein (ZBP1-Long) and a short isoform that contains the two Zα domains but lacks the RHIMs (R). (B) UCSC Genome browser snapshot of the *Zbp1* gene (173,206,612–173,218,923) on chromosome 2. Single-cell RNA sequencing data displayed as genome coverage from the Tabula Muris project (Tabula Muris et al, 2018). (C–E) Immunoblot of ZBP1 in total lysates of thymus and spleen (C), IFNα (1,000 U/ml)-treated lung fibroblasts (D), and BMDMs (E) from WT and *Zbp1⁻/⁻* mice. TBP, GAPDH, and VINCULIN serve as loading controls. Data information: Data in (D) are representative of two independent experiments. Source data are available online for this figure.

(Fig. 2A,B). Both *Zbp1^{L/L}* and *Zbp1^{S/S}* mice were born at the Mendelian ratio, developed normally to adulthood, and did not show overt abnormalities, demonstrating that exclusive expression of the long or short isoforms did not cause spontaneous pathology.

To confirm that *Zbp1^{L/L}* and *Zbp1^{S/S}* mice expressed exclusively the long and short ZBP1 isoforms, respectively, we assessed ZBP1 expression in primary LFs by immunoblotting. LFs from wild-type mice produced both ZBP1-L and ZBP1-S upon stimulation with IFNα for 24 h, while only faint expression of ZBP1-L was detected in unstimulated cells (Fig. 2C). LFs from *Zbp1^{L/L}* mice expressed exclusively ZBP1-L while LFs from *Zbp1^{S/S}* mice expressed exclusively ZBP1-S, confirming that the genetically engineered loci produced the predicted isoforms (Fig. 2C). Interestingly, expression of the respective ZBP1 isoforms seemed to be slightly elevated in LFs of the *Zbp1^{L/L}* and *Zbp1^{S/S}* knock-in mice compared to wild-type

mice (Fig. 2C), suggesting that all transcriptional activity is directed towards the expression of the remaining isoform in the mutant cells, while this is split between the two isoforms in wild type cells.

After confirming the exclusive expression of ZBP1-S and ZBP1-L, we next sought to delineate the function of the two isoforms in ZBP1-mediated signaling. We showed previously that inhibition of caspase-8 triggers ZBP1-mediated necroptosis in cells that are pre-stimulated with IFNs to induce ZBP1 expression (Jiao et al, 2020). To assess the role of the two isoforms in ZBP1-mediated cell death, we stimulated primary LFs from WT, *Zbp1^{L/L}*, *Zbp1^{S/S}* and *Zbp1⁻/⁻* mice with IFNα for 24 h followed by treatment with the caspase inhibitor Emricasan (em) in the presence of etanercept (et) to block necroptosis induced by autocrine TNF signaling. Assessment of cell death by real-time imaging revealed that *Zbp1^{L/L}* LFs were highly sensitive to necroptosis induced by IFNα+em+et, while wild-type cells expressing both

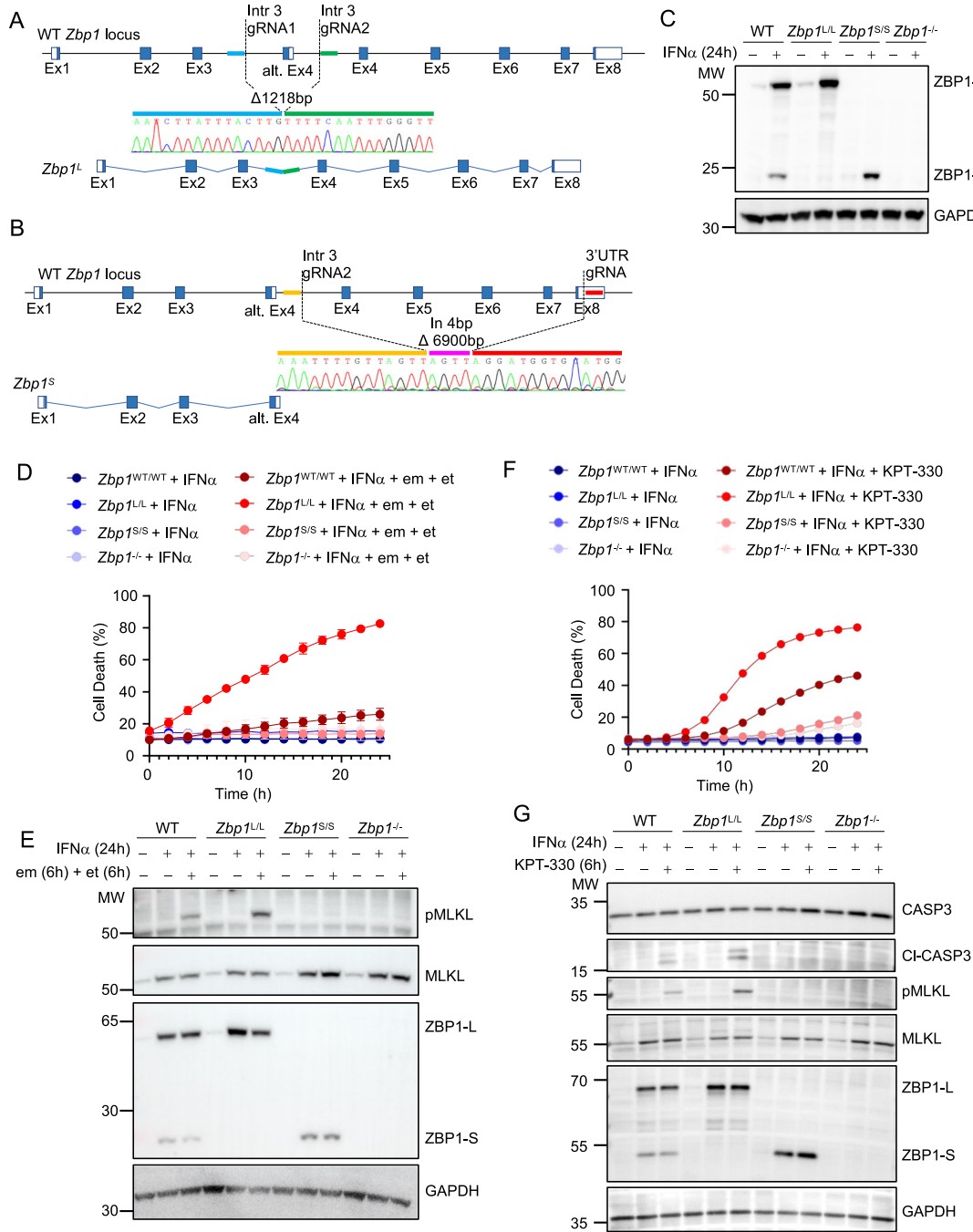

**Figure 2. Cells lacking endogenous ZBP1-S undergo increased ZBP1-L-induced cell death.**

(A, B) Schematic depicting the generation of mice expressing only the long (*ZBP1^L*) (A) or short isoform of ZBP1 (*ZBP1^S*) (B) using CRISPR-Cas9-mediated gene targeting in C57BL/6 N zygotes, as indicated. To generate *Zbp1^{L/L}* mice, alternative exon 4 (alt. Ex4) was removed. To generate *Zbp1^{S/S}* mice, the region including exons 4 to 8 was removed. The resulting deletion (1218 bp for *Zbp1^L* and 6900 bp for *Zbp1^S*) was confirmed by sequencing the new fusion site, and for *Zbp1^S* an additional 4 bp insertion was detected. (C) Immunoblot analysis of total lysates of lung fibroblasts from WT, *Zbp1^{L/L}*, *Zbp1^{S/S}*, and *Zbp1^{−/−}* mice treated with IFNα (1000 U/ml) for 24 h. (D, E). Cell death measured by DRAQ7 uptake (D) and immunoblot analysis of total lysates (E) of lung fibroblasts from WT, *Zbp1^{L/L}*, *Zbp1^{S/S}* and *Zbp1^{−/−}* mice stimulated with IFNα (1000 U/ml) (24-h pretreatment) alone or in combination with Emricasan (em) (2.5 μM) and etanercept (et) (50 μg/ml). (F, G) Cell death measured by DRAQ7 uptake (F) and immunoblot analysis of total lysates (G) of lung fibroblasts from WT, *Zbp1^{L/L}*, *Zbp1^{S/S}*, and *Zbp1^{−/−}* mice stimulated with IFNα (1000 U/ml) (24-h pretreatment) alone or in combination with KPT-330 (10 μM). Graphs show the percentage of cell death normalized to the total cell number obtained by 0.1% Triton X-100-induced cell lysis. All values are mean ± SEM from triplicate wells for each genotype (*n* = 3). In immunoblot analyses, GAPDH was used as a loading control. Data information: Data were representative of 3 (C–G) independent experiments. Source data are available online for this figure.

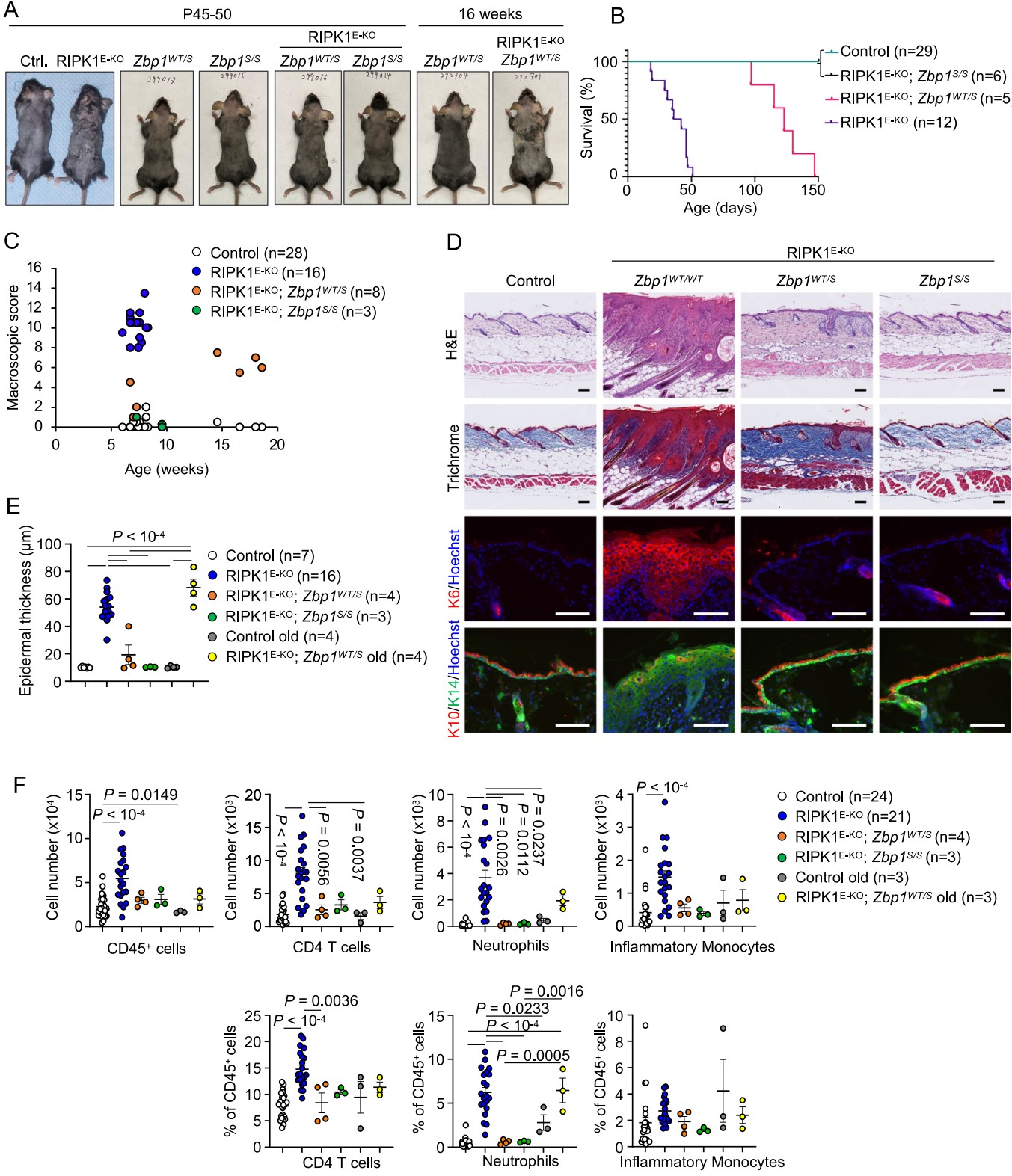

Figure 3.   Specific ablation of ZBP1-L prevents keratinocyte necroptosis and skin inflammation in RIPK1^E-KO mice.

(A) Representative macroscopic mouse pictures from mice with the indicated age and genotypes. (B) Kaplan–Meier plot depicting the time of sacrifice due to lesion severity of mice with the indicated genotypes. (C) Macroscopic skin score of mice with the indicated age and genotypes. (D) Representative images of skin sections from 6–10-week-old mice with the indicated genotypes, stained with haematoxylin and eosin (H&E), Trichrome, anti-keratin 6 (K6) antibodies, and Hoechst (DNA stain) or anti-keratin 10 (K10), anti-keratin 14 (K14) antibodies and Hoechst. Histological pictures shown are representative of control ($n = 7$), RIPK1^E-KO $Zbp1^{WT/WT}$ ($n = 16$ for H&E and Trichrome), ($n = 10$ for K6 and K10/K14); RIPK1^EKO $Zbp1^{WT/S}$ ($n = 4$), RIPK1^EKO $Zbp1^{S/S}$ ($n = 3$). Scale bars, 100 μm. (E) Microscopic quantification of epidermal thickness on the skin sections from 6–10- or 14–19 (old)-week-old mice with the indicated genotypes. (F) Flow cytometric quantification of CD45^+ cells, CD4^+ T cells (CD45^+CD3^+CD4^+), neutrophils (CD45^+CD3^-NK1.1^-CD19^-CD11b^+Ly6G^+), and inflammatory monocytes (CD45^+CD3^-NK1.1^-CD19^-Ly6G^-CD11b^+Ly6C^+) in skin from 6–10 or 14–19 (old) weeks old mice with indicated genotypes. Data information: Control mice include $Ripk1^{fl/fl}$ that do not express K14-Cre or $Ripk1^{fl/wt}$ K14-Cre mice, with wild type or $Zbp1^S$ alleles. Dots represent individual mice. Mean ± SEM is shown. $P$ values were calculated by one-way ANOVA with Tukey´s multiple comparisons test. Source data are available online for this figure.

isoforms showed very small numbers of dying cells under the same conditions (Fig. 2D). Consistent with the cell death assay results, immunoblot analysis revealed increased phosphorylation of MLKL in $Zbp1^{L/L}$ cells compared to WT cells (Fig. 2E). $Zbp1^{S/S}$ LFs were resistant to IFNα+em+et-induced necroptosis similarly to $Zbp1^{-/-}$ cells (Fig. 2D,E). Therefore, the absence of ZBP1-S sensitized cells to ZBP1-L-induced necroptosis, suggesting an inhibitory role of the short isoform. In our previous work, we discovered that treatment of cells with nuclear export inhibitors caused ZBP1-mediated cell death by inducing RIPK3-MLKL-dependent necroptosis and FADD-Caspase-8-dependent apoptosis (Jiao et al, 2020). To assess the role of the short and long isoforms on ZBP1-mediated cell death induced by inhibition of nuclear export, we treated primary LFs from wild type, $Zbp1^{L/L}$, $Zbp1^{S/S}$ and $Zbp1^{-/-}$ mice with IFNα for 24 h followed by incubation with the nuclear export inhibitor KPT-330. As shown in Fig. 2F, KPT-330 treatment induced cell death in WT LFs that was inhibited in $Zbp1^{-/-}$ and $Zbp1^{S/S}$ cells. Importantly, LFs from $Zbp1^{L/L}$ mice showed strongly increased cell death compared to WT cells after KPT-330 treatment (Fig. 2F). Moreover, immunoblot analysis of total protein extracts revealed increased MLKL phosphorylation as well as caspase-3 cleavage in $Zbp1^{L/L}$ compared to WT LFs after stimulation with IFNα +KPT-330 (Fig. 2G), showing that ZBP1-L induced stronger activation of necroptosis and apoptosis in the absence of the short isoform. Taken together, these results showed that $Zbp1^{S/S}$ cells were resistant to ZBP1-mediated cell death similarly to $Zbp1^{-/-}$ cells, while $Zbp1^{L/L}$ cells showed increased ZBP1-mediated cell death compared to WT cells expressing both isoforms. Therefore, ZBP1-S is unable to induce cell death but acts as an inhibitor suppressing cell death induced by the full-length ZBP1-L isoform.

## Loss of expression of ZBP1-L prevents skin inflammation in RIPK1^E-KO mice

To investigate how the absence of ZBP1-L and exclusive expression of ZBP1-S affected ZBP1-mediated cell death and inflammation in vivo, we bred $Zbp1^{S/S}$ mice with RIPK1^E-KO animals that develop skin inflammation induced by ZBP1-mediated keratinocyte necroptosis (Dannappel et al, 2014; Lin et al, 2016). Consistent with our cell culture experiments, RIPK1^E-KO $Zbp1^{S/S}$ mice did not develop skin lesions showing that loss of expression of ZBP1-L prevented ZBP1-mediated keratinocyte necroptosis and skin inflammation in vivo (Fig. 3A–C). Notably, heterozygous loss of ZBP1-L expression in RIPK1^E-KO $Zbp1^{WT/S}$ mice considerably delayed the development of skin lesions compared to RIPK1^E-KO animals. Specifically, while RIPK1^E-KO mice developed severe inflammatory skin lesions reaching the ethical endpoint requiring their humane sacrifice between 3 and 8 weeks of age, RIPK1^E-KO

$Zbp1^{WT/S}$ mice reached similar lesion severity between 17 and 22 weeks of age (Fig. 3A–C). Histological analysis of skin sections revealed severe inflammatory skin lesions in 6–8-week-old RIPK1^E-KO mice, characterized by epidermal hyperplasia, increased expression of keratin 6 (K6) and K14 and decreased expression of K10 (Fig. 3D). At this age, RIPK1^E-KO $Zbp1^{S/S}$ mice showed normal skin histology, while the skin of RIPK1^E-KO $Zbp1^{WT/S}$ mice showed only very small focal areas with mildly increased epidermal thickness (Fig. 3D,E). Flow cytometric assessment of immune cell populations revealed increased numbers of infiltrating CD45^+ cells, including CD4^+ T cells, neutrophils (Ly6G^+CD11b^+) and inflammatory monocytes (Ly6C^+CD11b^+) in the skin of 6–8-week-old RIPK1^E-KO mice, which were prevented by homozygous or heterozygous expression of ZBP1-S (Figs. 3F and EV2). When sacrificed at the age of 17–22 weeks, RIPK1^E-KO $Zbp1^{WT/S}$ mice showed increased epidermal thickness and mildly elevated numbers of infiltrating neutrophils in the skin, consistent with the macroscopic detection of skin lesions (Fig. 3A–C,E,F). Collectively, these results showed that loss of expression of the full-length ZBP1-L isoform prevented ZBP1-mediated keratinocyte necroptosis and inflammation in vivo in RIPK1^E-KO mice, further supporting that the short isoform is unable to induce ZBP1-mediated cell death.

## Loss of ZBP1-S exacerbates skin inflammation in RIPK1^E-KO mice

$Zbp1^{L/L}$ cells showed enhanced ZBP1-mediated cell death in vitro, revealing that the short isoform acts as an inhibitor of full-length ZBP1. To assess how the loss of ZBP1-S expression affects ZBP1-L-mediated necroptosis and inflammation in vivo, we crossed RIPK1^E-KO with $Zbp1^{L/L}$ mice. Strikingly, we found that homozygous or heterozygous loss of ZBP1-S expression dramatically accelerated the development of skin lesions in RIPK1^E-KO mice. Specifically, RIPK1^E-KO $Zbp1^{L/L}$ but also RIPK1^E-KO $Zbp1^{WT/L}$ mice developed severe skin lesions requiring their humane sacrifice already 8–12 days after birth, as opposed to RIPK1^E-KO mice that showed only minimal focal epidermal thickening at this age and reached the ethical endpoint between 3 and 7 weeks of age (Fig. 4A–C). Histological analysis confirmed the presence of severe skin inflammation characterized by epidermal hyperplasia and altered keratin expression in both RIPK1^E-KO $Zbp1^{L/L}$ and RIPK1^E-KO $Zbp1^{WT/L}$ mice at the age of 8–10 days, in contrast to RIPK1^E-KO mice that showed only very mild focal lesions at this age (Fig. 4D,E). Flow cytometric assessment of immune cell populations revealed increased numbers of infiltrating CD45^+ cells, including CD4^+ T cells, neutrophils (Ly6G^+CD11b^+) and inflammatory monocytes

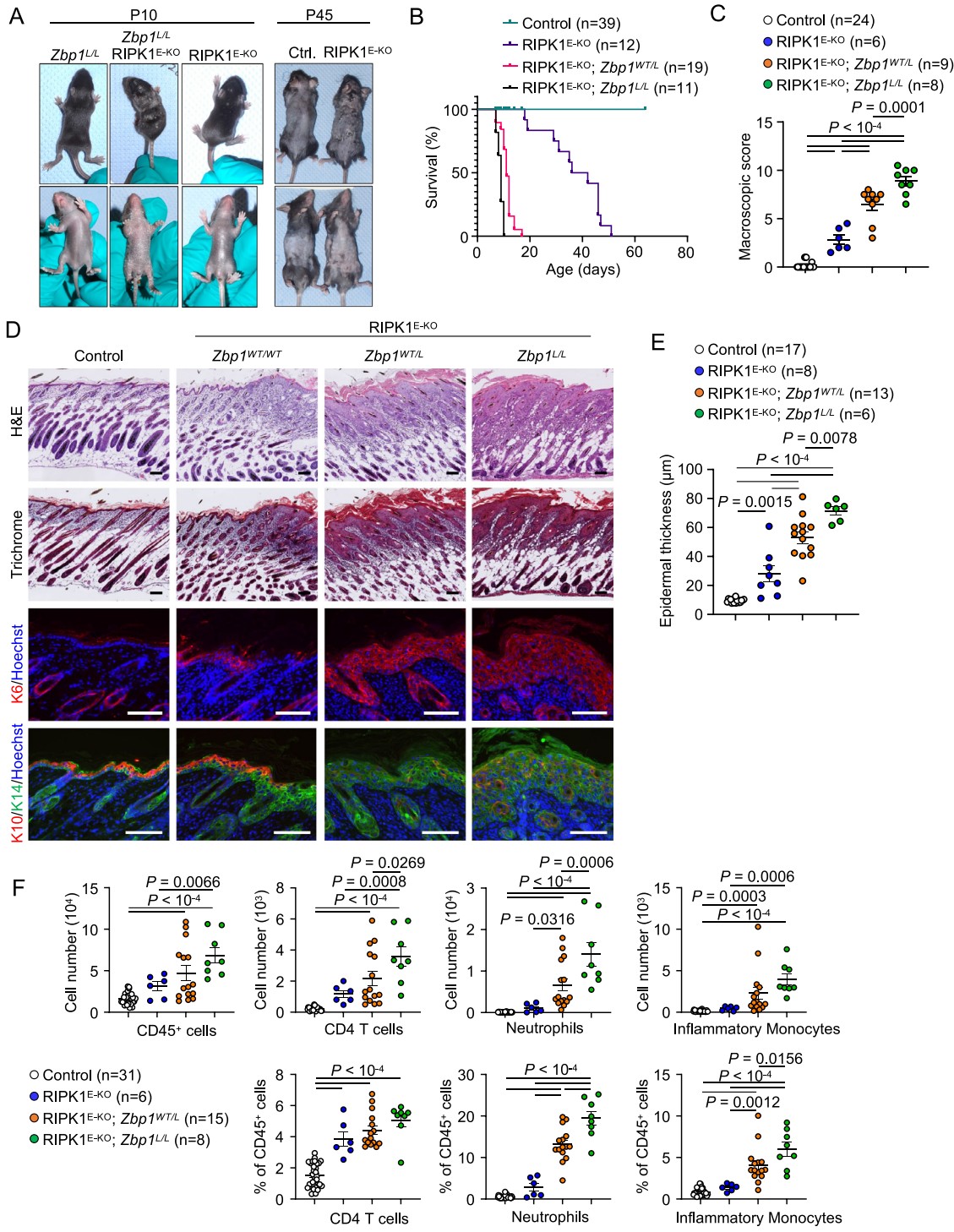

(Ly6C+CD11b+) in the skin of 8–10-day-old RIPK1[E-KO] Zbp1[L/L] and RIPK1[E-KO] Zbp1[WT/L] mice, as opposed to a very mild increase of CD4 T cells and neutrophils observed in RIPK1[E-KO] mice at this age (Fig. 4F). Thus, loss of ZBP1-S expression exacerbated ZBP1-L-induced necroptosis and inflammation in RIPK1[E-KO] mice, revealing a critical physiological function of the short isoform acting as an inhibitor to antagonize the activation of full-length ZBP1 in vivo.

## ZBP1-S inhibits ZBP1-L activation in a dose-dependent manner

Our findings described above revealed an important role of ZBP1-S in counteracting ZBP1-L-mediated cell death and inflammation. To interrogate the underlying mechanisms, we generated a cellular system that allows doxycycline (Dox)-inducible expression of fluorescently labeled ZBP1-S, ZBP1-L, and ZBP1mZα-L with

**Figure 4. Specific ablation of ZBP1-S accelerates and exaggerates ZBP1-L-induced skin inflammation in RIPK1^E-KO mice.**

(A) Representative macroscopic mouse pictures from mice with the indicated age and genotype. The images of the control and RIPK1^E-KO mice at P45 are the same as in Fig. 3A and are included here for comparison. (B) Kaplan–Meier plot depicting the time of sacrifice due to lesion severity of mice with the indicated genotype. The same group of RIPK1^E-KO mice as in Fig. 3B is included here for comparison. (C) Macroscopic skin scores of 1–2-week-old mice with the indicated genotypes. (D) Representative images of skin sections from 1–2-week-old mice with the indicated genotypes, stained with haematoxylin and eosin (H&E), Trichrome, anti-keratin 6 (K6) antibodies and Hoechst or anti-keratin 10 (K10), anti-keratin 14 (K14) antibodies and Hoechst. Histological pictures shown are representative of control (n = 24), RIPK1^E-KO Zbp1^WT/WT (n = 7), RIPK1^E-KO Zbp1^WT/L (n = 13), RIPK1^E-KO Zbp1^L/L (n = 6). Scale bars, 100 μm. (E) Microscopic quantification of epidermal thickness on the skin sections from 1–2-week-old mice with the indicated genotypes. (F) Flow cytometric quantification of CD45^+ cells, CD4^+ T cells (CD45^+CD3^+CD4^+), neutrophils (CD45^+CD3^-NK1.1^-CD19^-CD11b^+Ly6G^+), and inflammatory monocytes (CD45^+CD3^-NK1.1^-CD19^-Ly6G^-CD11b^+Ly6C^+) in the skin from 1–2-week-old mice with indicated genotypes. Data information: Control mice include Ripk1^fl/fl that do not express K14-Cre, or Ripk1^fl/wt K14-Cre mice, with wild type or Zbp1^L alleles. Dots represent individual mice. Mean ± SEM is shown. P values were calculated by one-way ANOVA with Tukey´s multiple comparisons test. Source data are available online for this figure.

mutations disrupting the capacity of the Zα domains to bind Z-nucleic acids (Z-NA). Specifically, we transduced Zbp1^−/− immortalized MEFs (iMEFs) with lentiviral constructs allowing Dox-inducible expression of ZBP1-L or ZBP1mZα-L fused to mNeonGreen (ZBP1-L-NG and ZBP1mZα-L-NG), or ZBP1-S fused to the red fluorescent protein mScarletI (ZBP1-S-Sc) (Fig. EV3A). To assess the role of the different isoforms, we generated Zbp1^−/− iMEFs expressing ZBP1-L-NG, ZBP1-S-Sc, ZBP1mZα-L-NG, or simultaneously expressing ZBP1-L-NG and ZBP1-S-Sc upon Dox treatment. In order to study how the expression level of ZBP1-S affects the activation of ZBP1-L, we used flow cytometry to sort two different populations of the cells expressing both the long and short isoforms, one expressing moderate levels of ZBP1-S-Sc and one expressing high levels of ZBP1-S-Sc after Dox treatment (Fig. EV3B). Stimulation with Dox +em induced cell death in iMEFs expressing ZBP1-L-NG but not in iMEFs expressing ZBP1mZα-L-NG (Fig. EV3C), consistent with our earlier studies showing that ZBP1-mediated cell death requires Zα-dependent sensing of endogenous Z-nucleic acids (Jiao et al, 2020). Moderate expression of ZBP1-S-Sc strongly suppressed while high ZBP1-S-Sc expression nearly completely prevented ZBP1-L-NG-induced cell death in response to Dox+em stimulation (Fig. EV3C), demonstrating that ZBP1-S inhibits the activation of ZBP1-L in a dose-dependent manner. Consistent with the cell death assays, immunoblot analysis revealed that expression of ZBP1-S-Sc in a dose-dependent manner suppressed RIPK3 and MLKL phosphorylation in cells expressing ZBP1-L-NG (Fig. EV3D). Assessment of ZBP1-dependent cell death induced by inhibition of nuclear export gave similar results. Incubation with Dox+KPT-330 induced cell death in iMEFs expressing ZBP1-L-NG but not ZBP1mZα-L-NG (Fig. EV3E,F), consistent with our earlier findings that nuclear export inhibition triggers Zα-dependent activation of ZBP1-mediated apoptosis and necroptosis (Jiao et al, 2020). Also, in response to the KPT-330 treatment, moderate expression of ZBP1-S-Sc partially inhibited while high expression almost completely suppressed ZBP1-L-NG-induced cell death (Fig. EV3E,F). Collectively, these results showed that ZBP1-S inhibited the activation of ZBP1-L in a dose-dependent manner.

## ZBP1-S requires functional Zα domains to suppress ZBP1-L activation

Next, we sought to understand how ZBP1-S restricted ZBP1-L-dependent cell death. ZBP1-S encompasses only the Zα domains and lacks the RHIMs, therefore it retains the capacity to bind Z-NA ligands but cannot engage RIPK1 or RIPK3 to induce downstream signaling. To assess whether Z-NA binding is required for ZBP1-S-mediated suppression of ZBP1-L-induced cell death, we generated iMEFs expressing Dox-inducible ZBP1-L-NG alone or together with cumate-inducible wild type ZBP1-S-Sc or Zα mutant ZBP1mZα-S-Sc (Fig. 5A,B). Incubation with Dox+em induced cell death in iMEFs expressing ZBP1-L-NG, which was considerably suppressed by cumate-inducible expression of ZBP1-S-Sc but not ZBP1mZα-S-Sc (Fig. 5C). Notably, iMEFs transduced with vectors expressing Dox-inducible ZBP1-L-NG + cumate-inducible ZBP1-S-Sc showed reduced cell death compared to iMEFs transduced with Dox-inducible ZBP1-L-NG + cumate-inducible ZBP1mZα-S-Sc in the absence of cumate, likely caused by basal expression of the short isoform due to leakage of the cumate-inducible promoter (Fig. 5B,C). Immunoblot analysis revealed that expression of ZBP1-S-Sc reduced ZBP1-L-NG-induced RIPK3 and MLKL phosphorylation but expression of ZBP1mZα-S-Sc had no effect (Fig. 5D), consistent with the cell death assay results. Therefore, the presence of functional Zα domains is required for ZBP1-S-mediated suppression of ZBP1-L-induced necroptosis in response to emricasan treatment. We then assessed if functional Zα domains are required for ZBP1-S-mediated suppression of ZBP1-L-induced cell death induced by nuclear export inhibition. Indeed, cumate-induced expression of ZBP1-S-Sc reduced ZBP1-L-NG-induced cell death after KPT-330 treatment, while expression of ZBP1mZα-S-Sc had no effect (Fig. 5E). In line with the cell death assays, immunoblot analysis showed that expression of ZBP1-S-Sc but not of ZBP1mZα-S-Sc suppressed ZBP1-L-NG-induced MLKL and RIPK3 phosphorylation as well as activation of caspase-8 and caspase-3 in KPT-330-treated cells (Fig. 5F). Taken together, these results showed that ZBP1-S inhibits the activation of ZBP1-L-mediated necroptosis and apoptosis in a manner that depends on functional Zα domains.

## ZBP1-S antagonises the activation of ZBP1-L by competing for binding to Z-RNA

Our results thus far showed that ZBP1-S antagonizes the activation of ZBP1-L in a Zα domain-dependent manner. To interrogate further the mechanism by which ZBP1-S counteracts ZBP1-L activation, we generated stable clones expressing ZBP1-L-NG, ZBP1mZα-L-NG, ZBP1-S-Sc, and both ZBP1-L-NG + ZBP1-S-Sc in a Dox-inducible manner, as well as ZBP1mZα-S-Sc in a cumate-inducible manner (Fig. 6A). Assessment of cell death in response to emricasan or KPT-330 confirmed our previous findings in bulk cell populations, showing that expression of ZBP1-S inhibited cell death induced by the long isoform (Fig. 6B). We considered whether

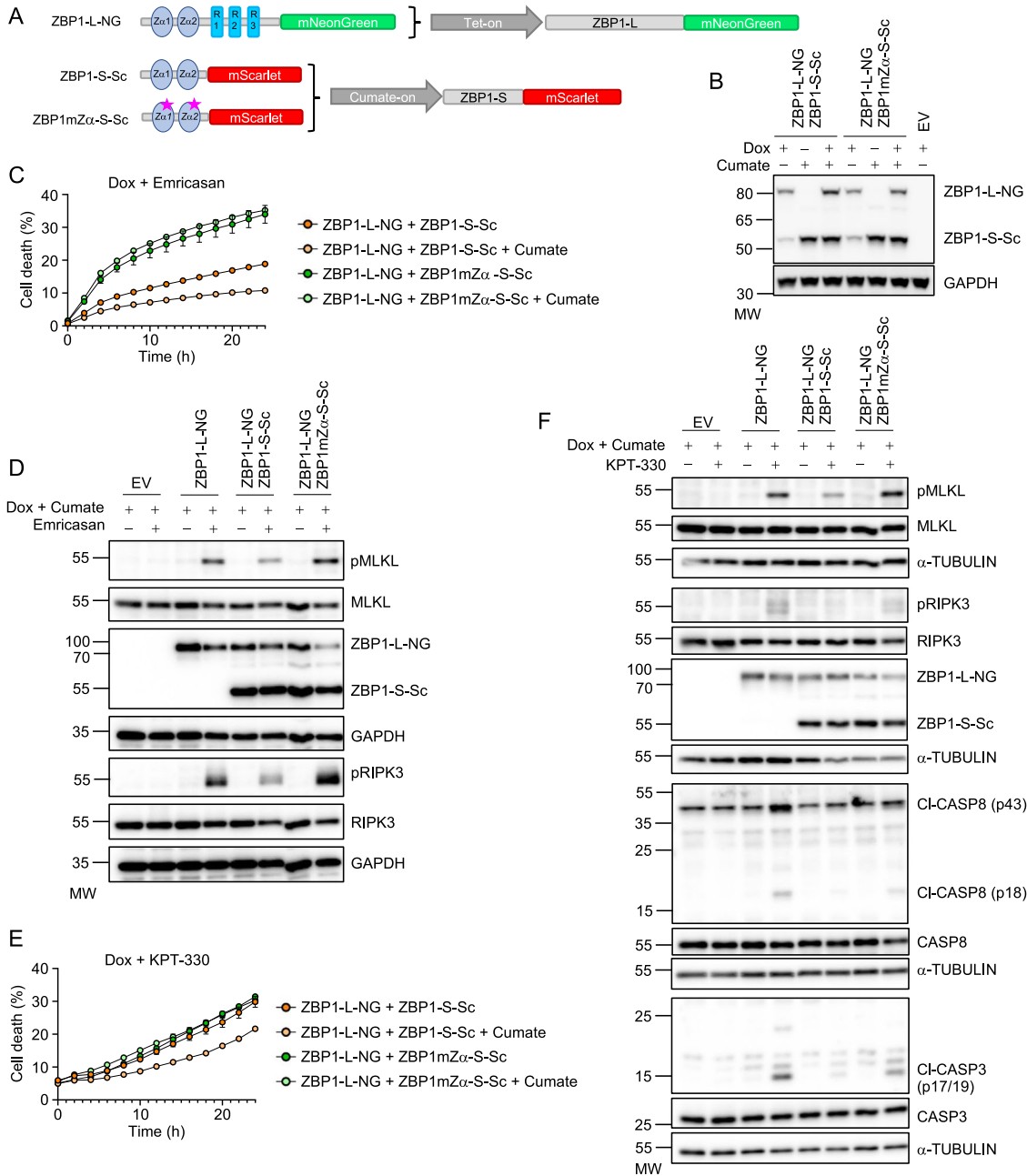

**Figure 5. Inducible expression of ZBP1-S suppresses ZBP1-L-mediated cell death in a Zα-dependent manner.**

(A) Schematic depicting the fusion proteins of ZBP1-L-NG, ZBP1-S-Sc, or ZBP1mZα-S-Sc and the combination of the Dox and cumate-inducible systems. (B) Immunoblot analysis of total lysates from iMEFs expressing doxycycline (dox)-induced empty vector (EV), the combination of dox-induced ZBP1-L-NG with ZBP1-S-Sc or ZBP1mZα-S-Sc stimulated with a combination of doxycycline (1 μg/ml) and cumate (100 μg/ml). (C, D) Cell death measured by DRAQ7 uptake (C) and immunoblot analysis of total lysates (D) in iMEFs expressing the indicated fusion proteins stimulated with combinations of doxycycline (1 μg/ml), cumate (50 μg/ml) (24 h pretreatment), and Emricasan (5 μM). (E, F). Cell death measured by DRAQ7 uptake (E) and immunoblot analysis of total lysates (F) in iMEFs expressing the indicated fusion proteins stimulated with combinations of doxycycline (1 μg/ml), cumate (50 μg/ml) (24 h pretreatment) and KPT-330 (10 μM). Cell death graphs show the percentage of cell death normalized to the total cell number obtained by 0.1% Triton X-100-induced cell lysis. Each value is presented as mean ± SEM from triplicate wells for each cell population ($n = 3$). For immunoblot analyses, GAPDH (B) and α-TUBULIN (D, F) were used as loading controls. Data information: Data were representative of 2 (B), 3 (C–F) independent experiments. Source data are available online for this figure.

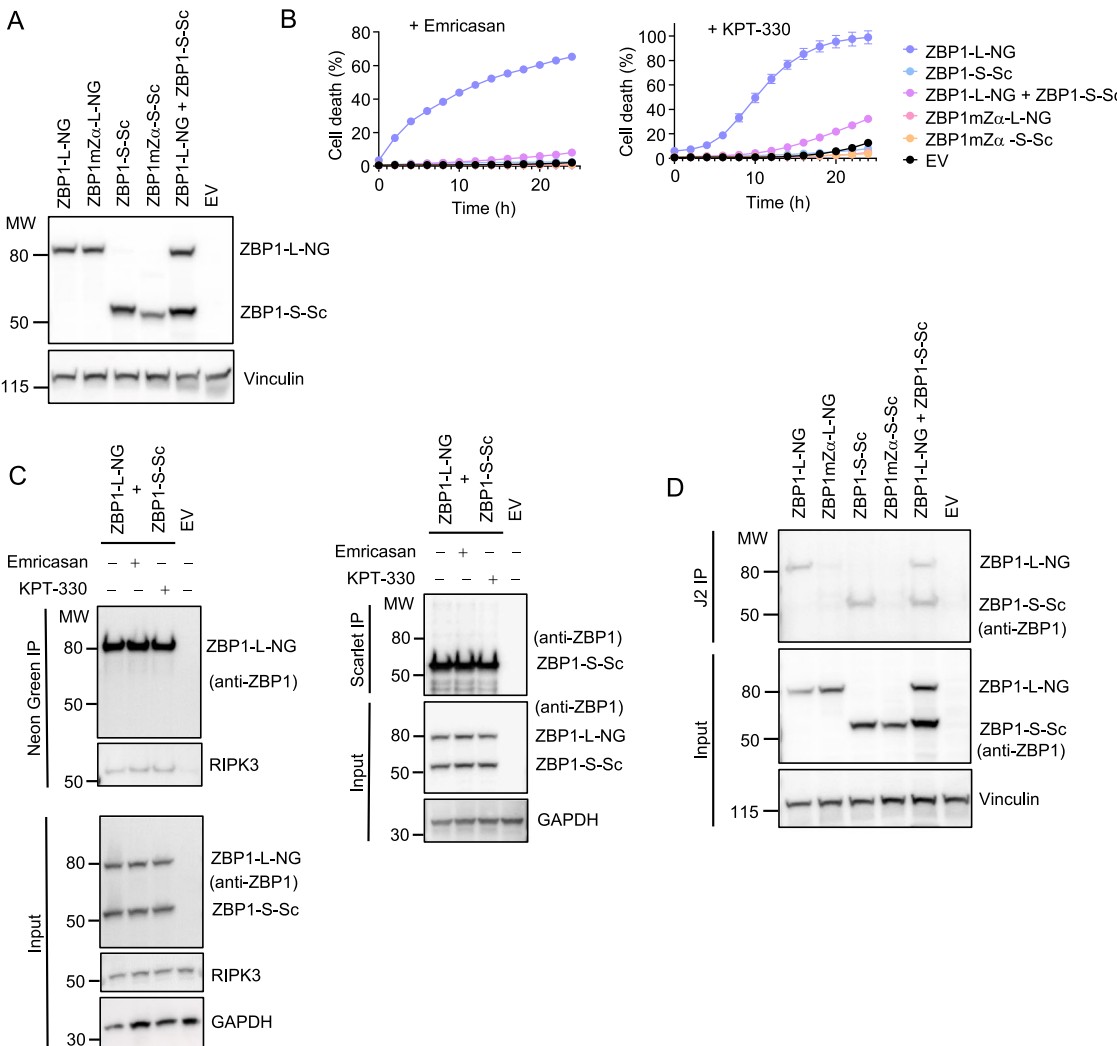

**Figure 6. ZBP1-L and ZBP1-S bind dsRNA via their Zα domains but do not interact with each other.**

(A) Immunoblot analysis of total lysates from iMEFs expressing doxycycline (dox)-induced ZBP1-L-NG, dox-induced ZBP1-S-Sc, the combination of dox-induced ZBP1-L-NG with ZBP1-S-Sc, dox-induced ZBP1mZα-L-NG,dox-induced empty vector (EV) stimulated with doxycycline (1 μg/ml) or cumate-induced ZBP1mZα-S-Sc stimulated with cumate (10 μg/ml) for 24 h. (B) Cell death measured by DRAQ7 uptake in iMEFs expressing the indicated fusion proteins stimulated with a combination of doxycycline (1 μg/ml) or cumate (10 μg/ml) (24 h pretreatment) and Emricasan (2.5 μM, left) or KPT-330 (10 μM, right). Graphs show the percentage of cell death normalized to the total cell number obtained by 0.1% Triton X-100-induced cell lysis. Each value is presented as mean ± SEM from triplicate wells for each cell population ($n = 3$). (C) Immunoblot analysis of anti-NeonGreen (left) and anti-mScarlet (right) immunoprecipitates and total lysates (input) from iMEFs expressing the indicated fusion proteins stimulated with doxycycline (1 μg/ml) alone for 24 h (D) or the combination of doxycycline (24 h pre-stimulation) with Emricasan (2.5 μM) or KPT-330 (10 μM) for 6 h. (D) Immunoblot analysis of J2 anti-dsRNA monoclonal antibody immunoprecipitates and total lysates (input) from iMEFs expressing indicated fusion proteins stimulated with doxycycline (1 μg/ml) for 24 h. GAPDH (C) and VINCULIN (A, D) serve as loading control. Data information: Data were representative of 2 (A), 3 (B–D) independent experiments. Source data are available online for this figure.

simultaneous binding of ZBP1-S and ZBP1-L to Z-nucleic acids might result in the formation of hetero-oligomers between ZBP1-S and ZBP1-L, where ZBP1-S might act to sterically hinder the RHIM-dependent interactions of ZBP1-L with other RHIM proteins. To assess this hypothesis, we immunoprecipitated ZBP1-L-NG with antibodies specific to mNeonGreen and assessed its interaction with ZBP1-S-Sc by immunoblotting with anti-ZBP1 antibodies. As shown in Fig. 6C, we did not detect ZBP1-S-Sc in the ZBP1-L-NG immunoprecipitates from cells that were either left untreated or treated with emricasan or KPT-330 to induce ZBP1-L-induced cell death. Consistently, ZBP1-S-Sc immunoprecipitation

with antibodies specific for mScarlet failed to pull down ZBP1-L-NG (Fig. 6C). These findings argue against a direct interaction as a mechanism of suppression of ZBP1-L by the short isoform. Alternatively, ZBP1-S might counteract ZBP1-L-induced cell death by competing for Z-NA ligands. To test this hypothesis, we assessed the capacity of the two isoforms to bind double-stranded RNA (dsRNA). As shown previously (Jiao et al, 2020), dsRNA immunoprecipitation with the monoclonal J2 antibody pulled down ZBP1-L-NG but not ZBP1mZα-L-NG (Fig. 6D), confirming that ZBP1 binds to dsRNA via its Zα domains. Immunoprecipitation of dsRNA from cells expressing both ZBP1 isoforms revealed

the presence of both ZBP1-L-NG and ZBP1-S-Sc in the J2 antibody immunoprecipitates (Fig. 6D), consistent with the presence of Zα domains in both isoforms. Some of our experiments indicated that ZBP1-S might be slightly enriched in dsRNA immunoprecipitates compared to the long isoform, but the data were not robust enough to draw firm conclusions (Fig. 6D). Whereas further studies will be needed to elucidate the detailed molecular mechanism by which the short isoform counteracts the activation of full-length ZBP1, these results are consistent with a model where ZBP1-S competes with ZBP1-L for binding to Z-RNA.

## Discussion

Here, we functionally characterized the two isoforms of ZBP1 expressed in mouse cells. The human ZBP1 gene exhibits a complex alternative splicing pattern, with at least 7 predicted isoforms annotated in Uniprot (Q9H171). One of these isoforms (Q9H171-5) contains only the first Zα domain and lacks RHIMs, thus resembling the mouse ZBP1-S. However, it remains unclear whether all of these isoforms are expressed at the protein level in human cells and tissues and whether they perform different functions. Our results revealed an important physiological function of the short isoform of mouse ZBP1 in suppressing cell death and inflammation induced by the full-length protein. Using both inducible expression in reconstituted cellular systems as well as primary cells from knock-in mice endogenously expressing exclusively each isoform, we showed that the short isoform inhibited ZBP1-L-induced necroptosis and apoptosis. The capacity of ZBP1-S to suppress ZBP1-L-induced cell death depended on the presence of functional Zα domains, demonstrating that binding of the short isoform to Z-NA ligands is required for antagonizing full-length ZBP1. Although the detailed molecular mechanism by which ZBP1-S inhibits the activation of the long isoform remains to be fully elucidated, our results suggest that the short isoform likely acts as a decoy competing with ZBP1-L for binding to cognate ligands. This mechanism is reminiscent of that of the vaccinia virus protein E3, which contains a Zα domain which binds Z-RNA to prevent the activation of ZBP1-mediated cell death that is induced in host cells as an antiviral mechanism (Koehler et al, 2021).

Notably, exclusive expression of ZBP1-L either using Dox-inducible over-expression or by IFN stimulation of the endogenous promoter was not sufficient to trigger substantial cell death even in the absence of the short isoform, suggesting that expression of ZBP1-S is not obligatory for preventing ZBP1-L activation and cell death induction. This is further supported by the finding that *Zbp1*^L/L^ mice expressing ZBP1-L but lacking the short isoform developed normally and did not show spontaneous pathology. These results are consistent with previous studies showing that ZBP1-mediated cell death is under tight negative regulation mediated by RIPK1 and caspase-8 (Devos et al, 2020; Jiao et al, 2020; Lin et al, 2016; Newton et al, 2016; Schwarzer et al, 2020; Solon et al, 2024; Yang et al, 2020). ZBP1 was also reported to induce NF-κB-dependent gene expression by engaging RIPK1 (Kaiser et al, 2008; Peng et al, 2022; Rebsamen et al, 2009), raising the question of whether ZBP1-S could also inhibit ZBP1-L-mediated NF-κB activation. However, in our previous experiments, we could not detect activation of NF-κB-dependent gene expression in response to doxycycline-inducible expression of ZBP1-L

(Koerner et al, 2024), therefore, we could not assess the possible role of ZBP1-S in regulating NF-κB signaling in our cellular system.

Our in vivo studies showing that loss of the short isoform strongly accelerated and exacerbated the development of inflammatory skin lesions in RIPK1^E-KO^ mice, which is induced by ZBP1-mediated necroptosis of RIPK1-deficient keratinocytes (Devos et al, 2020; Jiao et al, 2020; Lin et al, 2016), provided experimental evidence that ZBP1-S exerts a physiologically important function that is necessary to limit ZBP1-L-mediated cell death and inflammation. The physiological relevance of this finding for ZBP1-mediated responses in wild type mice remains to be investigated in future studies, however, we hypothesize that fine-tuning ZBP1-mediated cell death by the short isoform might be important to ensure efficient antiviral host defense while limiting tissue damage and pathology.

The two isoforms are co-expressed in different cell types in response to interferon-stimulation and are expressed at similar levels in mouse tissues including thymus and spleen. The co-regulation of the two isoforms suggests that their expression is controlled by the transcriptional activity of the common promoter, with the two different mRNAs generated by alternative splicing occurring stochastically at a constant rate. Our findings that the short isoform inhibits the activation of full-length ZBP1 are consistent with a recent study showing that expression of ZBP1-S suppresses ZBP1-L-mediated cell death in vitro in cellular systems (Cai et al, 2024). In that study, the authors report that the short isoform exhibits a shorter protein half-life but enhanced mRNA inducibility by IFN stimulation compared to the long one, suggesting that regulation of ZBP1-S expression at the level of protein stability may be relevant for its function. In our experiments, the short isoform mRNA was induced at slightly higher levels compared to the long one in response to IFN stimulation. Moreover, we detected a slightly reduced half-life for the ZBP1-S protein compared to ZBP1-L, but these differences were marginal and not as striking as reported by Cai et al, possibly due to different experimental conditions.

Our experiments provided evidence that ZBP1-S suppresses the activation of ZBP1-L in a dose-dependent manner. This is supported by using inducible ZBP1-S expression in cellular systems, where the expression level of ZBP1-S correlated with the extent of inhibition of ZBP1-L-mediated cell death. An even more striking demonstration of the dose-dependent function of ZBP1-S was revealed by our results that heterozygous expression of the short or the long isoform profoundly affected the kinetics and the severity of skin inflammation in RIPK1^E-KO^ mice. Specifically, RIPK1^E-KO^ *Zbp1*^WT/L^ mice, which express ZBP1-L from both alleles and ZBP1-S from only one allele, showed strongly accelerated and exacerbated skin inflammation compared to RIPK1^E-KO^ mice expressing both ZBP1 isoforms. Moreover, RIPK1^E-KO^ *Zbp1*^WT/S^ mice, which express the short isoform from both alleles but ZBP1-L from only one allele, showed strongly delayed and ameliorated skin inflammation compared to RIPK1^E-KO^ mice. These findings demonstrate that the ratio between the two isoforms profoundly affects ZBP1-mediated cell death and inflammation. Taken together, our results revealed an important role of the alternatively-spliced short isoform of ZBP1 in acting as an endogenous inhibitor of the full-length protein to fine-tune ZBP1-mediated cell death and inflammatory responses.

# Methods

## Mice

*Ripk1^{fl/fl}* (Dannappel et al, 2014) and *K14-cre* (Hafner et al, 2004) mice were described previously. *Zbp1* mutant mice were generated in the C57BL6/N genetic background using CRISPR/Cas9-mediated gene targeting in zygotes. To generate *Zbp1^{−/−}* mice, a short-guide (sg)RNA (Intr 1 gRNA: 5′-GAACGACGACACCACC-CAG-3′) upstream of Zα1-encoding exon 2 and a sgRNA (3′UTR gRNA: 5′-GTACATGTAACACCAACCAG-3′) targeting the 3′-UTR within exon 8 were in vitro-transcribed and co-injected into fertilized wild-type oocytes together with Cas9 protein and mRNA. To generate *Zbp1^{L/L}* mice, a sgRNA (Intr 3 gRNA1: 5′-TGAA-GATGCCGAGATCTGTG-3′) targeting the intron between exon 3 and the alternative exon 4 and a sgRNA (Intr 3 gRNA2: 5′-GCACACGTCTCCATTTGGAG-3′) targeting the intron between the alternative exon 4 and the canonical exon 4 were co-injected into fertilized wild-type oocytes with Cas9 protein and mRNA. To generate *Zbp1^{S/S}* mice, a sgRNA (Intr 3 gRNA1: 5′- GCA-CACGTCTCCATTTGGAG-3′) targeting the intron between the alternative exon 4 and the canonical exon 4 and a sgRNA (3′UTR gRNA: 5′-GTACATGTAACACCAACCAG-3′) targeting the 3′-UTR within exon 8 were co-injected into fertilized wild-type oocytes with Cas9 protein and mRNA. After confirmation of the correct mutations by genomic DNA sequencing analysis, founder mice carrying the targeted mutations were backcrossed to C57BL/6N mice to establish independent mouse lines. Mice were maintained at the specific-pathogen-free (SPF) animal facility of the CECAD Research Center of the University of Cologne. All mouse procedures were conducted in accordance with national and institutional guidelines, and protocols were approved by the responsible local authorities in Germany (Landesamt für Natur, Umwelt und Verbraucherschutz Nordrhein-Westfalen, license numbers: 84-02.04.2016.A452 and 81-02.04.2022.A298). Mice requiring medical attention were provided with appropriate care and sacrificed when reaching predetermined criteria of disease severity to minimize suffering. No other exclusion criteria existed. Calculations to determine group sizes were not performed. Mice of the indicated genotype were assigned at random to groups, and experiments were not blinded. Both male and female mice are included in all groups, as the phenotypes studied were not affected by the sex of the mouse.

## Immunoblotting and immunoprecipitation

Protein extracts from each tissue were prepared by homogenizing with beadmill (Precellys 24) in RIPA buffer supplemented with cOmplete mini protease inhibitor cocktail (04693124001, Roche) and phosSTOP phosphatase inhibitor tablets (4906837001, Roche) and denatured in 1 x Laemmli buffer (1610737 (2 x)/1610747 (4 x), Bio-Rad) supplemented with 10 mM DTT. Cell lysates were prepared by direct cell lysis and denaturation by 2 x Laemmli buffer supplemented with 5% β-mercaptoethanol and 10 mM DTT, or lysed on ice with a buffer containing 20 mM Tris pH 7.4, 150 mM NaCl, 2 mM EDTA, and 1% Triton X-100, supplemented with cOmplete mini protease inhibitor cocktail and phosphatase inhibitor phosSTOP tablets, that was cleared by centrifugation at 14,000 rpm for 20 min at 4 °C.

For the immunoprecipitation assays, iMEFs were plated in 15 cm dishes at a density of $6 \times 10^6$ cells per dish. *Zbp1*-gene expression was induced via a 24 h pretreatment with 1 µg/ml doxycycline (Sigma Aldrich, D9891-1G). The following day, the cells were collected and lysed in a lysis buffer containing 20 mM Tris pH 7.4, 150 mM NaCl, 2 mM EDTA, and 1% Triton X-100, supplemented with cOmplete mini protease inhibitor cocktail, phosphatase inhibitor phosSTOP tablets and 100 U/ml RNase inhibitor (M0314S, NEB). For the co-immunoprecipitation of dsRNA-interacting protein complexes, the cell lysates were incubated with anti-J2 antibody (RRID: AB_2651015) overnight at 4 °C on a rotator to allow the formation of antibody-antigen complexes. The next day, the complexes were captured by adding prewashed protein G Dynabeads (10004D, Life Technologies) to the samples and further incubating for 3 h at 4 °C under continuous rotation. Each sample was incubated with 2 µg of J2 antibody and 20 µg beads. For the NeonGreen-IP, the cell lysates were incubated with mNeonGreen-Trap Magnetic Agarose (ntma, ChromoTek), and for Scarlet-IP with RFP-Trap Magnetic Agarose beads (rtma, Chromotek) for 1 h at 4 °C on a rotator. The beads were washed four times with the same immunoprecipitation buffer, and the complexes were eluted from the beads by boiling them for 10 min in 4 × protein loading buffer.

Lysates were separated by SDS-PAGE, transferred to Immunobilon-P PVDF membranes (05317, Millipore) and, after blocking with 5% Skim milk powder (42590.02, SERVA) for 1 h, analysed by immunoblotting with primary antibodies against ZBP1 (RRID: AB_2490191), p-MLKL (RRID: AB_2799112), MLKL (RRID: AB_2820284), p-RIPK3 (RRID: AB_2799526), RIPK3 (RRID: AB_2039527), cleaved caspase-8 (RRID: AB_10891784), caspase-8 (RRID: AB_10545768), cleaved caspase-3 (RRID: AB_2341188), caspase-3 (RRID: AB_331439), GAPDH (RRID: AB_10077627), α-TUBULIN (RRID: AB_477582), VINCULIN (RRID: AB_2728768). Horseradish peroxidase (HRP)-conjugated secondary antibodies against rat (RRID: AB_2338128), rabbit (NA934V, Amersham Pharmacia) or mouse (NA931V, Amersham Pharmacia) were used for the detection of proteins using chemiluminescence with ECL SuperSignal West PicoPlus Chemiluminescent Substrate (34578, Thermo Fisher Scientific) or SuperSignal West Femto Maximum Sensitivity Substrate (34095, Thermo Fisher Scientific) and the signal was captured using a Fusion Solo X system (Vilber). Alternatively, the Odyssey system (Li-Cor) was used to detect the signal of fluorescently probed membranes. For fluorescent western blots, the Immobilon-FL PVDF membrane (IPFL00010, Millipore) and fluorescently labeled secondary antibodies IRDye 800CW Goat anti-Mouse IgG (RRID: AB_621842) and IRDye 680RD Goat anti-Rabbit IgG (RRID: AB_10956166) were used.

## Vector generation and ligase-independent cloning

Lentiviral vectors were generated using a combination of fusion PCRs and ligase-independent cloning. Previously generated vectors containing a single flag-tagged mouse ZBP1 coding sequence based on pCW-Cas9 (#50661, Addgene) were used to generate the required vectors for the expression of fluorescent fusion proteins (Jiao et al, 2020). Briefly, pCW-Cas9 had been linearized using NheI and BamHI. Full-length mouse *Zbp1* coding sequence was amplified from WT or *Zbp1^{mZa1-2}* (Jiao et al, 2020) cDNA using

zbp1-flag-fwd and zbp1-flag-rev to create amplicons with 20 bp homologies on both sides to the linearized vector. The resulting amplicons were inserted into the linearized pCW-backbone using ligase-independent cloning. Briefly, 100 ng PCR product and 100 ng vector were digested with 50U Exonuclease III (M0206, NEB) in 1xNEBuffer 1 for 1 min at RT, followed by column purification (T1030, NEB). Purified DNA was transformed into competent NEB Stable bacteria (C3040, NEB). The resulting pCW-Flag-ZBP1wt and pCW-Flag-ZBP1mZa1-2 plasmids were sequence-verified (Eurofins Genomics).

For pCW-Puro-WT-ZBP1L-mNeongreen, pCW-Puro-mZα-ZBP1L-mNeongreen, and pCW-Puro-WT-ZBP1S-mScarletI, the pCW-Flag-ZBP1WT plasmid was linearized using EcoRI and BamHI. WT and Zα1–2 mutant *Zbp1* full-length coding sequences were amplified using flag-zbp1L-mNG-fwd and flag-zbp1L-mNG-rev as well as for *Zbp1* short using primers flag-zbp1S-mS-fwd and flag-zbp1S-mS-rev. mNeongreen coding sequences were amplified using primers mNG-Fwd and mNG-Rev from 4xmts-mNeonGreen plasmid (#98876, Addgene), and the mScarletI sequence was amplified using primers mS-Fwd and mS-Rev from Lamp1-mScarletI (#98827, Addgene). ZBP1-fluorescence protein amplicons were generated using fusion PCR with the aforementioned amplicons. I.e., the ZBP1 amplicon was mixed in a 1:1 (w/w) ratio with the fusion protein amplicon and was amplified for 15 cycles using Q5 polymerase (M0494, NEB) followed by the addition of the ZBP1 fwd and the fluorescent protein rev primer and 30 additional cycles with Q5 polymerase. The resulting fusion amplicons were inserted into the linearized pCW-backbone using ligase-independent cloning to generate pCW-Puro-WT-ZBP1L-mNeongreen, pCW-Puro-mZα-ZBP1L-mNeongreen, and pCW-Puro-WT-ZBP1S-mScarletI.

For cumate inducible lentiviral vectors, first pCW-Cas9-Blast (#83481, Addgene) was digested with XhoI and EcorV. Then the cPPR/CTS fragment was amplified from the original pCW-Cas9-Blast using primers pcw-cuo-zbp1short-mscarlet-blast-F1 and pcw-cuo-zbp1short-mscarlet-blast-R1 as fragment 1. The cumate-sensitive promoter was amplified from PB-CuO-V5 CASC3 siRes 1–480 WT (#158541, Addgene) using primers pcw-cuo-zbp1short-mscarlet-blast-F2 and pcw-cuo-zbp1short-mscarlet-blast-R2 generating fragment 2. *Zbp1* short mScarletI fusion was amplified using primers pcw-cuo-zbp1short-mscarlet-blast-F3 and pcw-cuo-zbp1short-mscarlet-blast-R3 from pCW-Puro-WT-ZBP1S-mScarletI creating fragment 3. hPGK promoter was amplified from pCW-Puro-WT-ZBP1S-mScarletI using primers pcw-cuo-zbp1short-mscarlet-blast-F4 and pcw-cuo-zbp1short-mscarlet-blast-R4 yielding fragment 4. The cumate repressor cassette (CymR) was amplified from PB-CuO-V5 CASC3 siRes 1–480 WT using primers pcw-cuo-zbp1short-mscarlet-blast-F5 and pcw-cuo-zbp1short-mscarlet-blast-R5 as fragment 5. The blasticidin resistance cassette was amplified from pCW-Cas9-Blast using primers pcw-cuo-zbp1short-mscarlet-blast-F6 and pcw-cuo-zbp1short-mscarlet-blast-R6 as fragment 6. Finally, these 6 fragments were fused to two amplicons using Q5-based fusion PCR (1 + 2 + 3 and 4 + 5 + 6) and inserted into the linearized backbone using ligase-independent cloning. The resulting pCW-Blast-CuO-WT-ZBP1S-mScarletI vector was further modified to generate the final pCW-Blast-Cuo-WT-ZBP1S-mScarlet3 and pCW-Blast-CuO-mZα-ZBP1S-mScarlet3. To this end, pCW-Blast-CuO-WT-ZBP1S-mScarletI was opened using EcorI and MluI. WT and mZα1–2

*Zbp1* short coding sequences were amplified using primers pcw-cuo-zbp1short-mscarlet3-blast-F1 and pcw-cuo-zbp1short-mscarlet3-blast-R1 and the mScarlet3 gene was amplified from pLifeAct-mScarlet3_N1 (#189767, Addgene) using primers pcw-cuo-zbp1short-mscarlet3-blast-F2 and pcw-cuo-zbp1short-mscarlet3-blast-R2. *Zbp1* short and mScarlet3 amplicons were then fused by PCR using the external primers pcw-cuo-zbp1short-mscarlet3-blast-F1 and pcw-cuo-zbp1short-mscarlet3-blast-R2 followed by ligase-independent cloning into the linearized pCW-CuO backbone. pCW-Cas9 was a gift from Eric Lander and David Sabatini (#50661, Addgene), pCW-Cas9-Blast was a gift from Mohan Babu (#83481, Addgene), PB-CuO-V5 CASC3 siRes 1–480 WT was a gift from Niels Gehring (#158541, Addgene) and the plasmids 4xmts-mNeonGreen, Lamp1-mScarletI and pLifeAct-mScarlet3_N1 were gifts from Dorus Gadella (#98876, #98827, #189767, Addgene). All primers are described in Table EV1.

## Cell line generation

For lentivirus production, HEK293T cells were transfected with 10 μg of the different pCW based lentiviral vectors mentioned before together with 5 μg of psPAX2 (Addgene, #12260) and 5 μg of pMD2.G (Addgene #12259). Plasmids were mixed with 500 μl of 0.25 M $CaCl_2$ and 500 μl of 2 x HBS and incubated for 5 min at RT. The solution was added to HEK293T cells, which were fed with a new medium prior to transfection. After 1 day, the medium was replaced with fresh medium, and after 2 and 3 days, the supernatant containing the virus was collected. For transduction of iMEFs, 200,000 cells were incubated with 1.5 ml fresh medium, 1.5 ml viral supernatant, and 8 μg/ml Polybrene (Sigma Aldrich, #H9268) for 24 h. After 1 day, the virus-containing medium was replaced with a new medium containing 2 μg/ml puromycin or 10 μg/ml blasticidin. After 2 days, selection medium was removed and cell pools were used for experiments.

## Cell death assays

LFs and iMEFs were maintained at 37 °C and 5% $CO_2$ in DMEM (41965-039, Thermo Fisher Scientific) supplemented with 10% FCS (Biosell), 1% penicillin/streptomycin (15140130, Thermo Fisher Scientific), 1% L-glutamine (25030-123, Thermo Fisher Scientific), and 1 mM sodium pyruvate (11360, Thermo Fisher Scientific). For cell death assays, cells were seeded in 96-well plates at a density of $7 \times 10^3$ cells (iMEFs) or $1 \times 10^4$ cells (LFs) per well and cultured with IFNα (1000 U/ml), doxycycline (1 μg/ml) or cumate (50 or 100 μg/ml) for 1 day. The next day, cells were stimulated with Emricasan (2.5 or 5 μM) or KPT-330 (10 μM) in the presence of 0.1 μM DRAQ7 (DR71000, Biostatus). Cells were imaged for the indicated duration of time in 2 h intervals using the IncuCyte SX5 live-cell imaging and analysis platform (Sartorius) in bright-field and near-infrared fluorescence (excitation 648–674 nm and emission 685–756 nm) mode. DRAQ7-positive cells were automatically counted as dead cells in three images per well, and counts were averaged using the Incucyte software package version 2023A. The percentage of cell death was calculated by normalizing the DRAQ7 count per timepoint to the total cell count at the end of the experiment. The total cell count at the end was determined by lysing the cells in 0.1% Triton X-100 and 0.1 M DRAQ7, followed by imaging in the IncuCyte. For mNeongreen and mScarlet

expressing cells, the green (excitation 453–485 nm and emission 494–533 nm) and orange (excitation 546–568 nm and emission 576–639 nm) channels were additionally recorded.

## Gene expression analysis by qRT–PCR

Total RNA was extracted with the NucleoSpin RNA kit (Macherey-Nagel, 740955.50), according to the manufacturer's instructions, followed by cDNA synthesis using the LunaScript RT SuperMix kit (New England Biolabs, E3010L). qRT–PCR was performed using the dye-based Luna Universal qPCR Master Mix (New England Biolabs, M3003X) in a QuantStudio 5 Real-Time PCR System (ABI). Reactions were run in technical duplicates with Tbp as a reference gene. Relative expression of gene transcripts is shown using the $2^{-\Delta\Delta Ct}$ method. All qPCR primers are indicated in Table EV1.

## Protein stabilization assay

LFs were seeded in 12-well plates at a density of $1 \times 10^5$ cells per well and cultured with IFNα (1000 U/ml) for 1 day. The next day, cells were stimulated with CHX (10 μg/ml) or MG132 (5 μM) for 6 or 12 h. Cell lysates were prepared by direct cell lysis and denaturation by 2 x Laemmli buffer supplemented with 5% β-mercaptoethanol and 10 mM DTT and analysed ZBP1 protein level with immunoblotting.

## Histological analysis

Skin samples from each mouse were fixed in 4% paraformaldehyde (PFA) and embedded in paraffin. Each sample was cut into 3–5 μm sections and subjected to histological analysis by hematoxylin and eosin, Masson's Trichrome staining or immunofluorescence (IF) analysis. IF was performed with anti-K14 (RRID: AB_306091, 1:200), anti-K6 (RRID: AB_2565052, 1:600) and anti-K10 (RRID: AB_2565049, 1:300) antibodies. Immunostainings were visualized with Alexa-488 (RRID: AB_143165) and Alexa-594 (RRID: AB_2534079) fluorescent-conjugated secondary antibody, and all sections were counterstained with 5 μM Hoechst 33342 (62249, Thermo Fisher). Bright-field images were acquired using the NanoZoomer S360 Digital slide scanner (Hamamatsu) and fluorescent images were acquired Revolve (Echo). Images were analysed and processed using the Omero software package (https://openmicroscopy.org) and NDP.view2 Viewing software (Hamamatsu).

## Macroscopic scoring of skin lesions

The mice were regularly monitored for skin lesion development and were given a severity score according to the following criteria:

 0 = no skin lesion
 1 = alopecia or excoriations, 1 or 2 small punctuate crusts
 2 = multiple, small crusts
 3 = coalescing crusts
 4 = coalescing crusts with open lesions
 5 = multiple open lesions

The final scoring value was represented by the sum of each area score (Back skin, head, hip and belly skin). Final scoring was performed with blinded mouse genotypes.

## Skin FACS analysis

Representative skin lesion areas (2–5 cm², back skin area) were used for FACS analysis. Each skin sample was chopped into small pieces and digested in 2 ml of Digestion Cocktail (300 μg/ml of Liberase TL (5401020001, Roche), 50 U/ml DNase I (04716728001, Roche) (in RPMI) at 37 °C for 1.5-2 h with shaking (2000 rpm). Some debris, including hair, were removed by a 70- or 100-μm cell strainer after the digestion. Single-cell suspensions were subjected to each staining step as follows:

1. Near-IR Dead Cell Stain Kit (L10119, Thermo Fisher) at RT for 30 min in PBS
2. FC blocking with anti-Mouse Cd16/CD32 antibody (RRID: AB_394657) on ice for 10 min in FACS buffer (4% FBS in PBS with 2 mM EDTA)
3. Staining by each antibody (Ly6C-PerCP-Cy5.5 (RRID: AB_1659242), CD4-AF488 (RRID: AB_389302), NK-1.1-AF647 (RRID: AB_2132713), MHC class II (I-A/I-E)-AF700 (RRID: AB_494009), CD45-Super Bright 702 (RRID: AB_2662424), CD3-BV785 (RRID: AB_2562554), γδ T-Cell Receptor-BV650 (RRID: AB_2738530), CD11b-BV421 (RRID: AB_2562904), CD8-BV510 (RRID: AB_2563057), CD19-Super Bright 600 (RRID: AB_2637308), CD11c-PE-Cy7 (RRID: AB_2033997), Ly6G-PE (RRID: AB_1186104), and CD207-PE/Dazzle 594 (RRID: AB_2876491)) on ice for 30 min in FACS buffer.

After staining, debris was removed again with a 30-μm filter (04-0042-2316, Sysmex), and cell populations were detected by flow cytometry (BD LSRFortessa Cell Analyzer, BD Biosciences) and analysed by FlowJo.

## Experiment study design

There was no randomization for these experiments. All groups of experiments were performed using same protocols and experimental conditions. Each in vivo analysis such as determining epidermis thickness or final scoring of skin severity was performed with blinded genotypes.

## Data analysis and statistics

All statistical analyses were performed with GraphPad Prism v10. Statistical significances were assessed with one-way ANOVA with Tukey´s multiple comparisons test. $P$ values <0.05 are described in each Figure.

# Data availability

This study includes no data deposited in external repositories.

The source data of this paper are collected in the following database record: biostudies:S-SCDT-10_1038-S44318-024-00238-7.

# Peer review information

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

## Acknowledgements

We thank J Kuth, C Uthoff-Hachenberg, E Stade, E Gareus, P Roggan, M Hahn, and J von Rhein for technical assistance. We also thank the CECAD Transgenic Core Facility for CRISPR/Cas9-assisted gene targeting in mice, the CECAD Imaging Facility for microscopy support as well as the CECAD FACS Facility for support with flow cytometry. Research reported in this publication was supported by funding from the European Research Council (ERC) under the European Union's Horizon 2020 research and innovation program (Grant Agreement No. 787826), and the Deutsche Forschungsgemeinschaft (DFG, German Research Foundation, projects SFB1403 (Project No. 414786233) and under Germany's Excellence Strategy–EXC 2030 CECAD (project no. 390661388) to MP. MN was supported by a postdoctoral fellowship from the Alexander von Humboldt foundation.

## Author contributions

**Masahiro Nagata:** Data curation; Formal analysis; Validation; Investigation; Methodology; Writing—review and editing. **Yasmin Carvalho Schäfer:** Data curation; Formal analysis; Validation; Investigation; Methodology; Writing—review and editing. **Laurens Wachsmuth:** Data curation; Formal analysis; Validation; Investigation; Methodology; Writing—review and editing. **Manolis Pasparakis:** Conceptualization; Supervision; Funding acquisition; Writing—original draft; Writing—review and editing.

## Funding

## Disclosure and competing interests statement

The authors declare no competing interests.

# Expanded View Figures

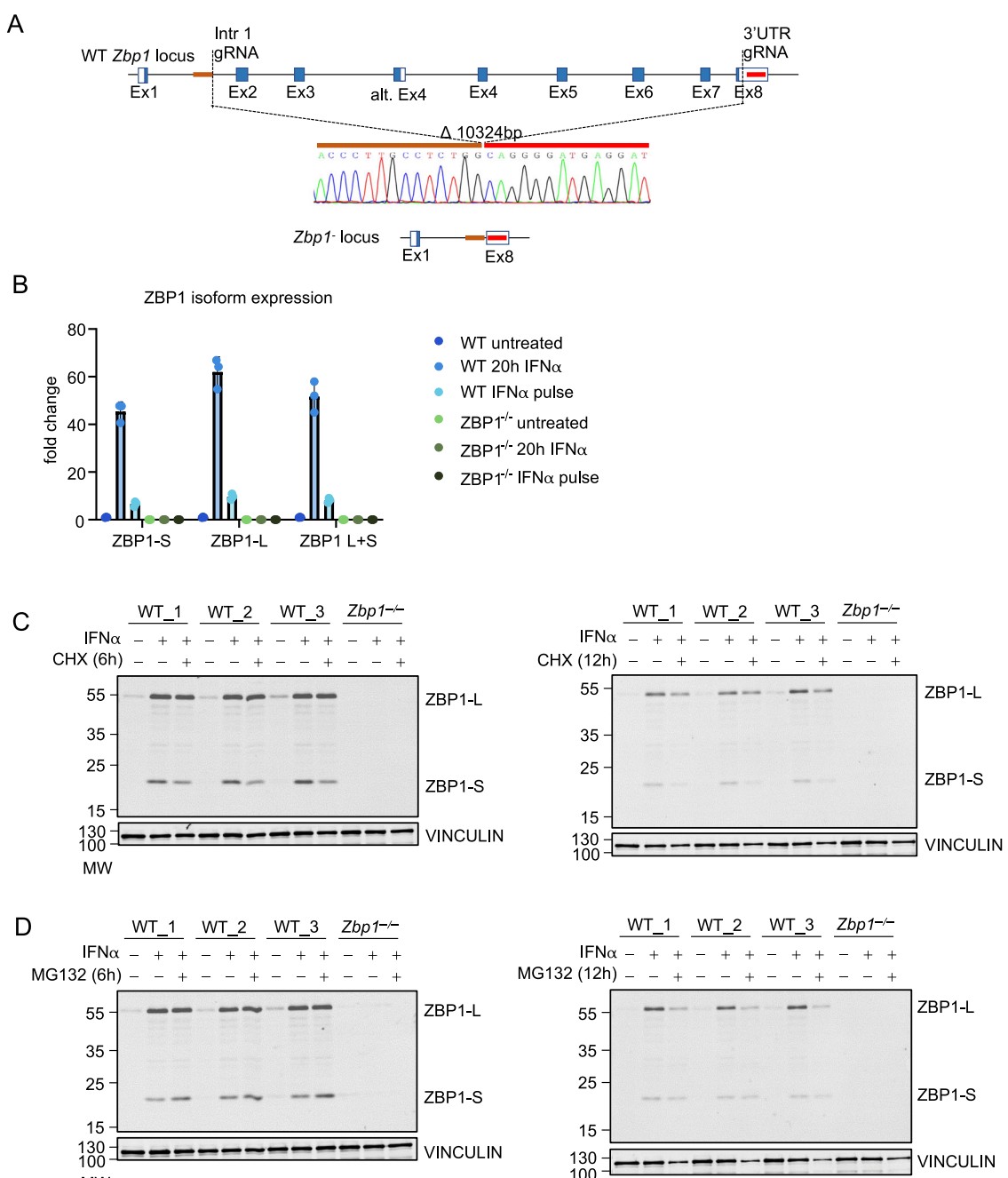

**Figure EV1.  Assessment of ZBP1-S and ZBP1-L mRNA and protein expression.**

(A) Schematic depicting the generation of novel ZBP1-deficient mice (*Zbp1*−/−) using CRISPR-Cas9-mediated gene targeting in C57BL/6N zygotes, as indicated. To generate *Zbp1*−/− mice, most of the *Zbp1* gene was removed by cutting once upstream of exon 2 and once within the 3′UTR in exon 8, leading to loss of the intervening region via nonhomologous end joining. The deletion event was confirmed by Sanger sequencing as indicated. (B) qPCR on RNA extracted from primary lung fibroblasts from three WT and three ZBP1−/− mice. Cells were either untreated, treated for 20 h with IFNα, or pulsed with IFNα (20 h IFNα treatment followed by washing and 24 h without IFNα), and primers were used to amplify only the short (ZBP1 Short), only the long (ZBP1 Long) or both (ZBP1 L + S) *Zbp1* isoforms. Data were presented as fold change relative to untreated WT samples. Dots represent individual mice. Mean ± SD is shown. (C, D). Immunoblot analysis with total lysates of lung fibroblasts from three WT mice and one *Zbp1*−/− mouse treated with IFNα (18 h pretreatment) and indicated chemicals (CHX, 10 μg/ml (C) or MG132, 5 μM (D)) for 6 or 12 h. Data information: Data were representative of 2 (B) independent experiments. Source data are available online for this figure.

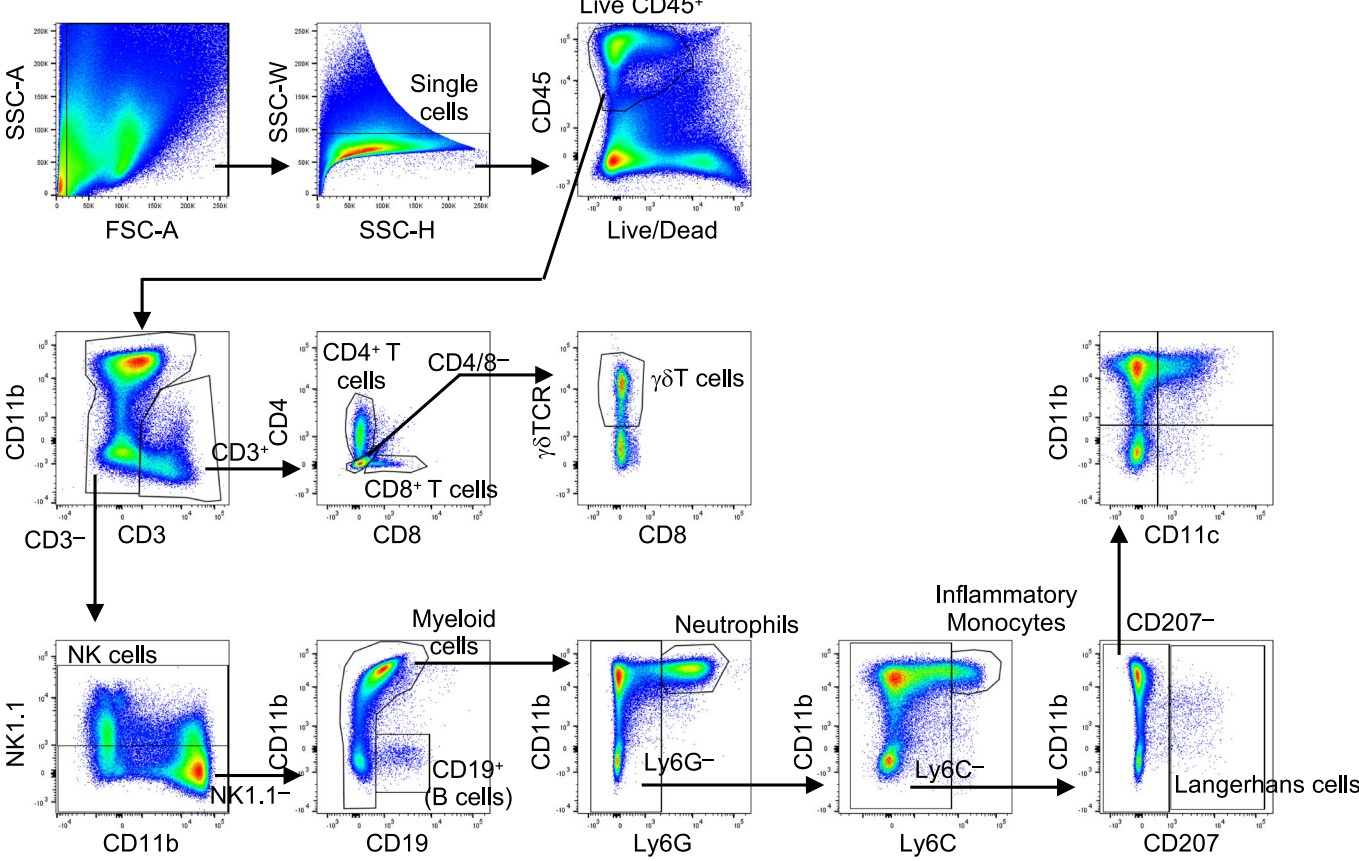

**Figure EV2.  Gating strategy for FACS analysis of immune cells in skin tissues.**

Gating strategy for analysing immune cells in digested skin with flow cytometry. The dot plots from RIPK1[E-KO] *Zbp1*[L/L] mouse is used as representative. The viability of cells is determined by the staining with LIVE/DEAD™ Fixable Near-IR Dead Cell Stain Kit (Live/Dead). Dead cells are presented as Live/Dead[+], and live cells are presented as Live/Dead[−]. FSC forward scatter, SSC side scatter.

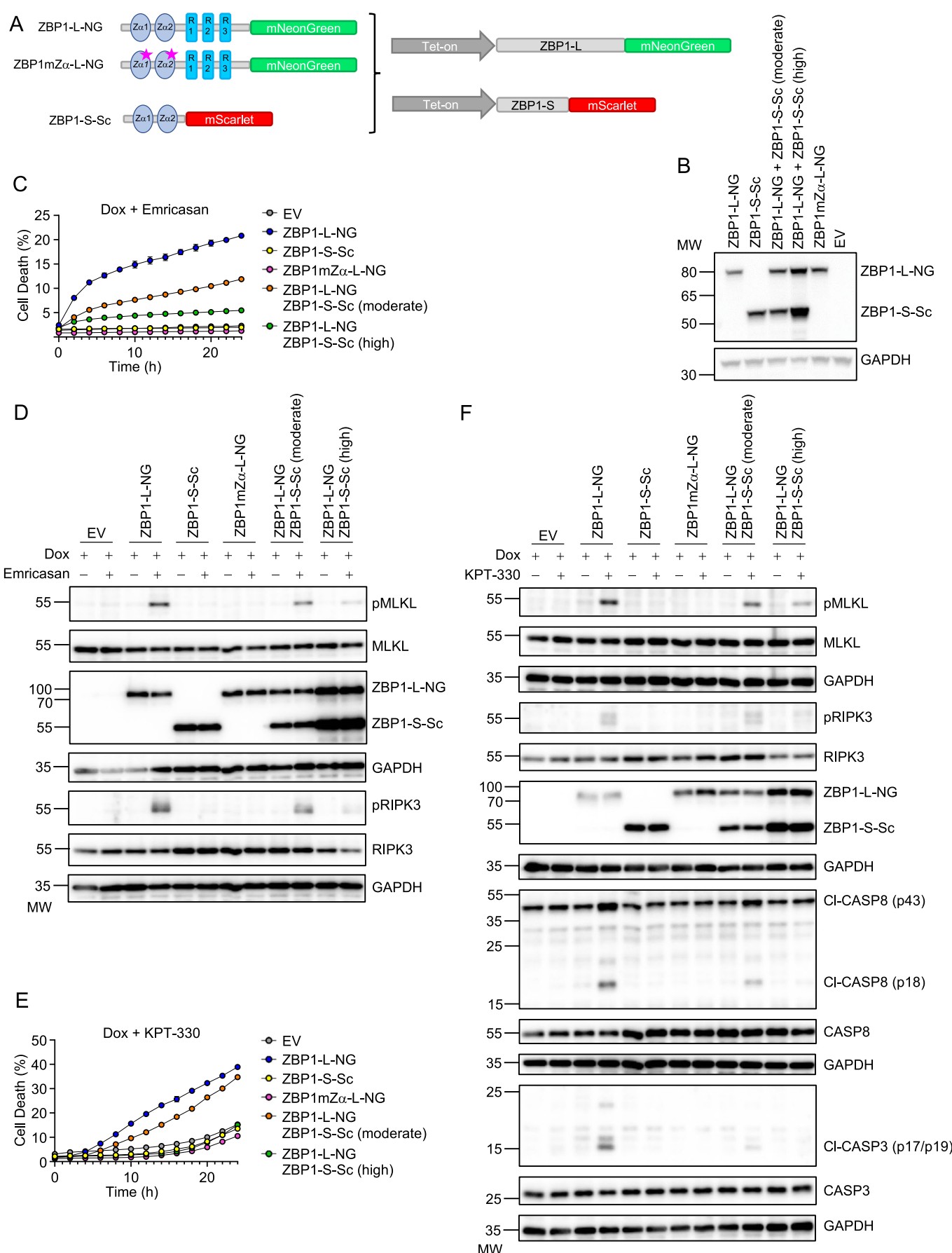

◀ **Figure EV3. ZBP1-S expression levels correlate with its capacity to suppress ZBP1-L-mediated cell death.**

(A) Schematic depicting the fusion proteins of ZBP1-L-NG, ZBP1mZα-L-NG, or ZBP1-S-Sc and the Dox-inducible systems. (B) Immunoblot analysis of total lysates from immortalized MEFs expressing doxycycline (dox)-inducible ZBP1-L-NG, ZBP1-S-Sc, ZBP1-L-NG with moderate or high expression levels of ZBP1-S-Sc, ZBP1mZα-L-NG, and empty vector (EV) stimulated with doxycycline for 24 h. (C, D) Cell death measured by DRAQ7 uptake (C) and immunoblot analysis of total lysates (D) in iMEFs expressing the indicated fusion proteins stimulated with doxycycline (1 μg/ml) (24 h pretreatment) and Emricasan (5 μM). (E, F) Cell death measured by DRAQ7 uptake (E) and immunoblot analysis of total lysates (F) in iMEFs expressing the indicated fusion proteins stimulated with doxycycline (1 μg/ml) (24 h pretreatment) and KPT-330 (10 μM). Cell death graphs show the percentage of cell death normalized to the total cell number obtained by 0.1% Triton X-100-induced cell lysis. Each value is presented as mean ± SEM from triplicate wells for each cell population ($n = 3$). For immunoblot analyses, α-TUBULIN (D) and GAPDH (F) were used as loading controls. Data information: Data were representative of 1 (B), 3 (C–E), 4 (F) independent experiments. Source data are available online for this figure.

