## [Peer Review File · The EMBO Journal]

A shorter splicing isoform antagonizes ZBP1 to modulate cell death and inflammatory responses

Manolis Pasparakis, Masahiro Nagata, Yasmin Carvalho, and Laurens Wachsmuth

Corresponding author(s): Manolis Pasparakis (pasparakis@uni-koeln.de)

Review Timeline:

Submission Date:	8th Apr 24
Editorial Decision:	29th May 24
Revision Received:	1st Jul 24
Editorial Decision:	31st Jul 24
Revision Received:	15th Aug 24
Accepted:	20th Aug 24

Editor: Ioannis Papaioannou

Transaction Report:

Dear Manoli,

Thank you again for submitting your manuscript EMBOJ-2024-117374 for consideration by The EMBO Journal, and for your patience during peer review. Your manuscript has been seen by four experts in the field, and we have received the full set of their comments, which I have already shared with you (included again below). I would also like to thank you for taking the time to discuss with me their reports and your provisional revision plan in our very informative and constructive meeting.

The referees have provided well-informed reports and overall recognize the solidity of the presented data as well as the novelty and significance of the findings. They also identify, however, certain limitations and provide a number of suggestions for strengthening the manuscript further. Given the referees' comments and recommendations, and your willingness to address the majority of their concerns, I would like to invite you to submit a revised version of the manuscript along with a detailed point-by-point response addressing all referees' comments.

I should add that it is EMBO Journal policy to allow only a single round of major revision, and acceptance of your manuscript will therefore depend on the completeness of your responses in this revised version. Please let me know if you have any questions or comments that you would like to discuss with me. We generally allow three months as standard revision time (August 28, 2024). As a matter of policy, competing manuscripts published during this period will not negatively impact our assessment of the conceptual advance presented by your study. However, we request that you contact us as soon as possible upon publication of any related work, to discuss how to proceed.

Thank you for the opportunity to consider your work for publication in The EMBO Journal. I look forward to your revision.

Best regards,

Ioannis

Instructions for preparing your revised manuscript

1. When you are ready to submit the revision, please upload:

- A Word file of the manuscript text (including legends of main Figures, EV Figures and Tables). Please make sure that changes are highlighted (or "tracked") to be clearly visible.

- Individual production-quality figure files (one file per figure). When assembling your figures, please refer to our figure preparation guidelines in order to ensure proper formatting and readability in print as well as on screen:

If the data shown in a figure are obtained from n {less than or equal to} 2, please use scatter plots showing the individual data points.

- i. the name of the statistical test used to generate error bars and P values
- ii. the number (n) of independent experiments (please specify technical or biological replicates) underlying each data point (discussion of statistical methodology can be reported in the Materials and Methods section, but figure legends should contain a basic description of n , P , and the test applied)
- iii. the nature of the bars and error bars (s.d., s.e.m.).

- A point-by-point response to the referees' comments, with a detailed description of the changes made (as a word file). All referees' concerns must be fully addressed and their suggestions taken on board. When preparing your letter of response to the referees' comments, please bear in mind that this will form part of the Review Process File and will therefore be available online to the community. Please note that you have the possibility to opt out of the transparent process at any stage prior to publication by letting the editorial office know (contact@embojournal.org); if you do opt out, the Review Process File link will point to the following statement: "No Review Process File is available with this article, as the authors have chosen not to make the review

process public in this case.". For more details on our Transparent Editorial Process, please visit our website:
<https://www.embopress.org/page/journal/14602075/authorguide#transparentprocess>

- Expanded View (EV) files (replacing Supplementary Information) that are collapsible/expandable online. A maximum of 5 EV Figures can be typeset. EV Figures should be cited as "Figure EV1, Figure EV2" etc. in the text, and their respective legends should be included in the manuscript file after the legends of regular figures. See detailed instructions regarding Expanded View files here:

- For the figures that you do NOT wish to display as Expanded View figures, they should be bundled together with their legends in a single PDF file called "Appendix", which should start with a short Table of Contents (including page numbers). Appendix figures should be referred to in the main text as: "Appendix Figure S1, Appendix Figure S2" etc. Please see detailed instructions here: <https://www.embopress.org/page/journal/14602075/authorguide#expandedview>

- A complete author checklist, which you can download from our author guidelines (<https://www.embopress.org/page/journal/14602075/authorguide>). Please note that the checklist will also be part of the Review Process File.

2. Please note that no statistics should be calculated and shown in Figures if $n=2$. Please also note that each p value should be reported as an exact value.

3. Before submitting your revision, primary datasets (and computer code, where appropriate) produced in this study need to be deposited in appropriate public databases (see <https://www.embopress.org/page/journal/14602075/authorguide#dataavailability>).

The accession numbers and databases should be listed in a formal "Data availability" section (placed after Materials and Methods) that follows the model below (see also <https://www.embopress.org/page/journal/14602075/authorguide#dataavailability>):

Data availability

- RNA-seq data: Gene Expression Omnibus GSE46843 (<https://www.ncbi.nlm.nih.gov/geo/query/acc.cgi?acc=GSE46843>)
- [data type]: [name of the resource] [accession number/identifier/doi] ([URL or identifiers.org/DATABASE:ACCESSION])

*** All links should resolve to a page where the data can be accessed. ***

*** Please remember to provide in the Data availability section of your revised manuscript reviewer passwords if the datasets are not yet public. ***

*** The Data Availability Section is restricted to new primary data that are part of this study. In case you have no data that require deposition in a public database, please state so: "Our study includes no data deposited in public repositories." under the heading "Data availability". ***

4. Please check that the title and the abstract of the manuscript are brief, yet explicit, even to non-specialists. The length of the title should not exceed 100 characters, and the abstract should be a single paragraph not exceeding 175 words.

5. Please also note our reference format: <https://www.embopress.org/page/journal/14602075/authorguide#referencesformat>.

7. Please remember: digital image enhancement is acceptable practice, as long as it accurately represents the original data and conforms to community standards. If a figure has been subjected to significant electronic manipulation, this must be noted in the figure legend or in the "Materials and Methods" section. The editors reserve the right to request original versions of figures and the original images that were used to assemble the figure.

8. Our journal encourages inclusion of data citations in the reference list to directly cite datasets that were obtained from public databases. Data citations in the article text are distinct from normal bibliographical citations and should directly link to the database records from which the data can be accessed. In the main text, data citations are formatted as follows: "Data ref: Smith et al, 2001" or "Data ref: NCBI Sequence Read Archive PRJNA342805, 2017". In the Reference list, data citations must be labeled with "[DATASET]". A data reference must provide the database name, accession number/identifiers, and a resolvable link to the landing page from which the data can be accessed at the end of the reference. Further instructions are available at:

<https://www.embopress.org/page/journal/14602075/authorguide#referencesformat>.

9. We request authors to consider both actual and perceived competing interests. Please review our policy (<https://www.embopress.org/page/journal/14602075/authorguide#conflictsofinterest>) and update your competing interests statement if necessary. Please name this section 'Disclosure and competing interests statement' and place it after the Acknowledgements section.

10. Please note that all corresponding authors are required to provide an ORCID ID upon submission of a revised manuscript (<https://orcid.org/>). Please find instructions on how to link your ORCID ID to your account in our manuscript tracking system in our Author guidelines (<https://www.embopress.org/page/journal/14602075/authorguide#authorshipguidelines>).

11. We use CRediT to specify the contributions of each author in the journal submission system. CRediT replaces the author contribution section, which should be removed from the manuscript. Please use the free text box to provide more detailed descriptions. See also guide to authors: <https://www.embopress.org/page/journal/14602075/authorguide#authorshipguidelines>.

13. We would also welcome the submission of cover suggestions or motifs to be used by our Graphics Illustrator in designing a cover.

14. Please use the link below to submit your revision:
<https://emboj.msubmit.net/cgi-bin/main.plex>

Referee #1:

ZBP1 is an important sensor in the innate immune system: it detects DNA/RNA in the unusual Z-form and induces regulated cell death and inflammation. ZBP1 thereby plays important roles in viral infections and in autoinflammatory diseases. Studies into the mechanisms of ZBP1 regulation are thus important and timely. Here, the authors characterise the role of a short splice isoform of mouse ZBP1 that lacks the signalling motifs (RHIMs). Using a combination of mouse genetics, in vitro and in vivo data, the authors conclude that the short ZBP1 isoform is an antagonist of the full-length protein. The data support this conclusion extremely well. I find this is an important result that should be published. The authors further suggest that the mechanism involves competition for RNA agonists; however, the data are somewhat preliminary. Another shortcoming of the manuscript is the lack of comparison of ZBP1 splice isoforms across species. Indeed, Ensemble does not annotate a similar splice isoform for human ZBP1. In conclusion, with some revision, this manuscript is a strong candidate for EMBO J.

Major Points

1. The authors should repeat the experiment shown in Fig 6D by including ZBP1mZa-S constructs. The ZBP1-L band in lane 3 appears weaker compared to lane 1. Is this reproducible across repeat experiments, and can this be quantified?
2. This reviewer is worried that a number of data/panels across multiple figures were not repeated in independent experiments (incl Fig 6D). At least two (and ideally three) repeats should be performed throughout, or a compelling reason for not repeating experiments provided.
3. The authors need to provide information about alternative splicing of human ZBP1 and discuss PMID 36755096 in this regard. Moreover, the authors should consider comparing ZBP1 splicing across multiple species.

Minor Points

1. Line 43. Please check reference formatting.
2. Lines 77/78. Why did the authors not cite PMID 34686350 here?
3. Line 166. Please explain further why TNF signalling needs to be blocked by et.
4. The y-axis labelling in Fig 3B and 4B is confusing. Survival should start at 100%, not reach 100% with time.
5. In Fig 4A, please double check the indicated genotypes. Is the animal on the left L/L or wt/wt for Zbp1?
6. Line 294: insert 'and' between caspase-3 and caspase-8

Referee #2:

In the study, the authors generated mouse models expressing exclusively either ZBP1-L (full length) or ZBP1-S (short isoform) endogenously. Using these mice and cells, the authors show that ZBP1-S regulates ZBP1-L function in vitro and in vivo. ZBP1-S restricts ZBP1-L-mediated cell death and inflammation-associated pathologies. The study and the mechanisms of ZBP1-S in regulating ZBP1-L are interesting. I have a few critical comments. Please see below.

1. The expression of the ZBP1-S in mice has been reported/discussed in previous studies (isoform-2 of the ZBP1), and authors have ignored this or were not aware of the existing literature (PMID: 32350114; PMID: 32296175, one of their own studies). It has been demonstrated that the deletion of the Za2 domain of ZBP1 abolishes ZBP1-S form expression. (PMID: 32350114).
2. Whether human cells express the ZBP1-S isoform has not been explored. This needs to be tested for biological relevance. Most of the human cell lines may not express ZBP1, and thus, the authors should consider isolating PBMCs (or human monocyte/macrophage cells) to test the ZBP1-L and ZBP1-S expression.
3. The mechanism shows that ZBP1-S antagonises the activation of ZBP1-L by competing for binding to Z-RNA. The only evidence provided was J2-antibody pull-downs. However, dsRNA pull down by J2-antibody does not provide direct evidence of ZBP1-S antagonizing ZBP1-L by restricting Z-RNA binding. The authors should perform pull-down experiments with Z-nucleic acid (Z-NA) specific antibody as used in multiple studies (Mouse monoclonal anti-Z-NA antibody, clone Z22) and probe for detection of ZBP1-S and ZBP1-L.
4. The authors did not cite the literature appropriately. In page-2, lines 77 & 81, the authors should cite the following reference (PMID: 32350114). This study generated mice specifically lacking Za2 domain and showed Z-RNA binding specific regulation of ZBP1 functions.
5. Page no-3, Line-126: Authors stated, "whereas we did not detect expression of either isoform in unstimulated BMDM." Authors should be cautious while writing such sentences. Previous studies show the basal expression of ZBP1 in unstimulated BMDMs. This sentence needs to be corrected.

Referee #3:

General summary and opinion about the principal significance of the study, its questions and findings:

Nagata, Shafer, Wachsmuth & Pasparakis use mouse genetics and cellular models to demonstrate that the stoichiometry between two isoforms of mouse ZBP1 regulates downstream apoptosis and necroptosis. The short mouse isoform, ZBP1-S, does not contain the two/three RHIMs that are present in the long mouse isoform and that are required to mediate signaling. The authors propose a competition model whereby ZBP1-S sequesters Z-nucleic acids without forming a complex with ZBP1-L. In general the study is solid; especially the mouse genetic studies are very convincing and well-controlled; i.e. data from ZBP1 knockout cells and heterozygous crosses contribute to the quality of the study. Importantly, this work also confirms an independent study now published in Cell Reports (PMID 38748877). I recommend to address the following concerns to solidify some conclusions related to the mechanism by which ZBP1-S inhibits ZBP1-L and to validate (or not) data from PMID 38748877.

Specific major concerns essential to be addressed to support the conclusions:

- Is ZBP1-S unique to mouse cells? If so, it is important to stress this point. I recommend to add this to the title, make it clearer in the abstract (i.e. on line 29-30 in the final sentence of the abstract) and throughout the main text. I find this an important point since it was not openly addressed in PMID 38748877.
- Does ZBP1-S have a shorter half-life compared to ZBP1-L as proposed in PMID 38748877? It would be important to test (e.g. using a simple CHX chase assay). Even if the results do not match, this should be reported.
- Human ZBP1 also induces NF- κ B activation, at least in an assay whereby the protein is inducibly expressed using a Tet-On system (PMID 36268590). The authors should refer to this work in the introduction (e.g. on lines 50-51). Does mouse ZBP1 similarly induce NF- κ B and does ZBP1-S regulate this process? The authors have developed the perfect system to test this idea. For example, RT-qPCR and/or ELISA for NF- κ B activated genes after addition of doxycycline would do the job.
- Figure 2D and related Western blot: why is there no detectable cell death in wild type cells after IFN- α /em+et stimulation? Is it a titration issue or the fact that IFN- γ (as described by the authors in PMID 32296175), but not IFN- α triggers ZBP1-dependent cell death in these cells? The authors should address this experimentally.
- Readers that are less familiar with ZBP1 signaling and how apoptosis vs necroptosis is regulated may be confused about the hierarchy of ZBP1 signaling and the choice for IFN- α /Em/Et (necroptosis) and KPT-330 (apoptosis + necroptosis) to activate ZBP1. As far as I know, it is not clear how the nuclear export inhibitor KPT-330 activates ZBP1. It would be more logical to test IFN- α /smac mimetic (e.g. BV6)/Et to try and induce ZBP1-mediated apoptosis in analogy with TNF signaling.
- Figure 6 is the weakest part of the study. Several controls are missing and should be included. Did the co-IP in figure 6C work? For instance, does RIPK3 co-IP under conditions of em or KPT-330? The reverse co-IP (i.e. ZBP1-S IP) should also be performed in these conditions. Can the authors include a control to show that the J2 IP worked (e.g. dot blotting for dsRNA). The authors state that ZBP1-S competes with ZBP1-L to bind to Z-RNA. Have the authors tried to IP Z-RNA/DNA using Z22? Moreover, from the figure legend, it appears that the experiment in figure 6D has only been performed once. This experiment should be repeated.

Minor concerns that should be addressed:

- Line 92-93: Karki et al 2021 (PMID 34686350) also report that ADAR1 inhibits ZBP1. It would be fair to also refer to this study.
- Line 125/Figure 1C/D and line 322/Figure 6D: Quantitative statements based on Western blots should be avoided. For example transfer times affect the ratios of larger vs smaller proteins on the blotting membrane.
- Line 195, figure 3B: Zbp1S/S rescued mice are not on the graph
- Please provide gating strategies for figures 3F and 4F
- Line 305: why would ZBP1-S form heterodimers with ZBP1-L? Could also be hetero-oligomers.
- Line 365: the preprint has now been published (PMID 38748877)
- Could the authors change the color schemes and/or change symbols in the cell death graphs? The different conditions are hard to discern.
- Figures 3B and 4B, the Y-axis should depict "dead (%)" instead of "survival (%)".
- Line 305-306: Confocal fluorescent microscopy can determine the localization of ZBP1-S and ZBP1-L

Any additional non-essential suggestions for improving the study (which will be at the author's/editor's discretion):

I wonder whether a more condensed form of the manuscript would be better. Figure 6 is thus far less substantiated compared to figure 5. Since both figures address the mechanism by which ZBP1-S suppresses ZBP1-L function both figures could be merged.

Referee #4:

In this study, Nagata et al. investigated the roles of the two ZBP1 isoforms, the short isoform (ZBP1-S) containing just the Z α domains, and the long isoform (ZBP1-L) containing the Z α domains and C-terminal RHIM domains. They show that ZBP1-S acts as an endogenous inhibitor of ZBP1-L. Using newly generated mouse models, the authors showed that loss of ZBP1-S exacerbated skin inflammation induced by ZBP1-mediated cell death in RIPK1E-KO mice. The study also used cellular systems to suggest that ZBP1-S inhibits ZBP1-L-induced cell death by competing for binding to Z-nucleic acids. Overall, these findings provide insights into the autoregulation of ZBP1. While another paper reporting similar findings regarding these two isoforms was recently published in Cell Reports (PMID: 38748877), the current study includes *in vivo* analyses, which extend beyond the Cell Reports study. The authors should perform additional analyses and consider alternative interpretations of their data to strengthen their study.

Major Comments

1. The authors observed higher expression of ZBP1-L when ZBP1-S was removed (Fig. 2C). This effect complicates the interpretation of cell death phenotypes. The authors should consider an alternative interpretation of their data: that the higher cell death observed might be due to increased ZBP1-L levels, with ZBP1-S controlling ZBP1-L expression. The authors should perform additional control experiments to test this possibility, such as designing transcript-specific primers or probes to test relative transcript levels of ZBP1-L and ZBP1-S.
2. In addition to its role in cell death, ZBP1 has cell death-independent, pro-inflammatory functions via RIPK1/3 (PMID: 36268590). The authors should determine the function of ZBP1-S in this context, including its effects on inflammatory gene expression in both fibroblast and myeloid cells.
3. To draw conclusions about the effect of ZBP1-S on inflammation specifically, the authors should test inflammatory cytokine release by ELISA from both the *in vitro* and *in vivo* studies.

Minor Comments

1. While the authors conclude that ZBP1-S may outcompete ZBP1-L for binding to nucleic acids, the data to support this conclusion in Figure 6D are not strong, and the mechanism through which this would happen is not clear. Is ZBP1-L completely dispensable for the interaction of ZBP1-S with nucleic acids? What are the consequences of ZBP1-L being partially pulled down along with the J2 antibody? To make this conclusion, the authors should assess the KD for each of the isoforms to interact with nucleic acids.
2. The authors should discuss the possible functions of ZBP1-S and ZBP1-L in different cancer models. For example, what could be the phenotype of these isoforms in tumor surveillance? Additionally, the authors should discuss the relevance of these findings for humans.
3. There is a regulatory relationship between ADAR1 and ZBP1. Assessing the correlation between ADAR1 expression and ZBP1-S may provide a broader view of ZBP1 regulation.
4. In the introduction, the authors note that ZBP1 mediates heatstroke-induced cell death; however, this is currently controversial in the field (PMID: 35511979, 38409108).

5. The survival graphs (Figure 3B and 4B) are difficult to follow, as the y-axis indicates it is % survival, but the actual data plotted seem to be % with disease or % dead.

6. Additional controls are needed in some of the western blots. For example, in Fig. 6D, a control for ZBP1mZ α -S is missing to show that ZBP1-S can bind dsRNA. Additionally, in Fig 2E and 2G, the total MLKL band is highly variable. The authors should explain this.

Point by point response to the reviewer comments

Referee #1

ZBP1 is an important sensor in the innate immune system: it detects DNA/RNA in the unusual Z-form and induces regulated cell death and inflammation. ZBP1 thereby plays important roles in viral infections and in autoinflammatory diseases. Studies into the mechanisms of ZBP1 regulation are thus important and timely. Here, the authors characterise the role of a short splice isoform of mouse ZBP1 that lacks the signalling motifs (RHIMs). Using a combination of mouse genetics, in vitro and in vivo data, the authors conclude that the short ZBP1 isoform is an antagonist of the full-length protein. The data support this conclusion extremely well. I find this is an important result that should be published. The authors further suggest that the mechanism involves competition for RNA agonists; however, the data are somewhat preliminary. Another shortcoming of the manuscript is the lack of comparison of ZBP1 splice isoforms across species. Indeed, Ensemble does not annotate a similar splice isoform for human ZBP1. In conclusion, with some revision, this manuscript is a strong candidate for EMBO J. We thank the reviewer for their thorough assessment of our manuscript and their insightful comments.

Major Points

1. The authors should repeat the experiment shown in Fig 6D by including ZBP1mZ α -S constructs. The ZBP1-L band in lane 3 appears weaker compared to lane 1. Is this reproducible across repeat experiments, and can this be quantified?

We have now repeated the J2 pulldown experiments including ZBP1mZ \$\alpha\$ -S constructs, which showed that mutation of the Z \$\alpha\$ domains prevent binding of ZBP1-S to dsRNA, as expected. The reviewer correctly points out that in the experiment included in this figure panel in the originally submitted manuscript it appeared as if co-expression of the short isoform might reduce binding of the long isoform to dsRNA. We have now repeated this experiment several times and, while we often see a bit stronger signal for ZBP1-S in the J2 IP, our overall conclusion is that these assays do not provide clear evidence that ZBP1-S expression reduces the binding of ZBP1-L to dsRNA. While we still favour the hypothesis that ZBP1-S suppresses ZBP1-L-induced cell death by competing for binding to Z-NAs, this experiment cannot provide a clear answer to this question. We have adjusted the text in the discussion to draw the readers' attention to this point. We have now replaced the blots in Fig 6D with new blots including also ZBP1mZ \$\alpha\$ -S, where ZBP1-S and ZBP1-L appear to bind with equal strength to dsRNA.

2. This reviewer is worried that a number of data/panels across multiple figures were not repeated in independent experiments (incl Fig 6D). At least two (and ideally three) repeats should be performed throughout, or a compelling reason for not repeating experiments provided.

We were of course aware of this issue, which was caused by the need to submit the paper as soon as possible because of the publication of the competing paper in BioRxiv. We have now repeated all experiments and updated the relevant information in the figure legends.

3. The authors need to provide information about alternative splicing of human ZBP1 and discuss PMID 36755096 <https://pubmed.ncbi.nlm.nih.gov/36755096/> in this regard. Moreover, the authors should consider comparing ZBP1 splicing across multiple species.

The role of ZBP1 in humans is a topic of intense investigation, but remains poorly understood. The human ZBP1 gene exhibits a complex alternative splicing pattern, with at least 7 predicted isoforms annotated in Uniprot (Q9H171). One of these isoforms (Q9H171-5) contains only the second Z \$\alpha\$ domain and lacks RHIMs, thus resembling the mouse ZBP1-S. However, it remains unclear whether all of these isoforms are expressed at the protein level in human cells and tissues and whether they perform different functions. This is largely due to the fact that it is very difficult to detect endogenous ZBP1 protein expression in most human cell types. We have included a short text at the beginning of the discussion section of our manuscript outlining the complex splicing pattern of human ZBP1. As the current literature on human ZBP1 is scarce and confusing, we opted not to discuss studies on human ZBP1 as we felt this would take too much space and might be counterproductive as we believe that for several of these studies it will be important to be reproduced across different labs before firm conclusions can be drawn.

The reviewer specifically refers to the study by Nassour et al, who showed that two main ZBP1 isoforms are expressed in primary human lung fibroblasts, which they named as ZBP1(L) and ZBP1(S). The isoform named ZBP1(S) by Nassour et al only lacks the first $Z\alpha$ domain and is fully functional as it contains the second $Z\alpha$ and all RHIMs, therefore, it is completely different from mouse ZBP1-S. It is unfortunate that this human isoform was named ZBP1 short as this could indeed confuse the readers when thinking about the mouse ZBP1 short protein. Overall, while we agree that studying the splicing pattern and functionally characterising the different isoforms in human cells, but also comparing and studying ZBP1 splicing across multiple species, will be very interesting and important, we respectfully suggest that this is outside the scope of the current manuscript which focuses on the functional characterisation of the two main isoforms in mice.

Minor Points

1. Line 43. Please check reference formatting.

We thank the reviewer for pointing out that there was a problem with the formatting of these references, which we have now corrected.

2. Lines 77/78. Why did the authors not cite PMID 34686350

<https://pubmed.ncbi.nlm.nih.gov/34686350/> here?

Will have now cited this reference.

3. Line 166. Please explain further why TNF signalling needs to be blocked by et.

We include the TNF blocker in the experiments because lung fibroblasts undergo TNF-mediated necroptosis when treated with caspase-8 inhibitors such as emricasan. This is caused by autocrine TNF signalling in these cultures and sometimes makes it difficult to accurately measure ZBP1-mediated necroptosis that happens at the same time. We have now clarified this in the text of the manuscript.

4. The y-axis labelling in Fig 3B and 4B is confusing. Survival should start at 100%, not reach 100% with time.

Indeed, the way these graphs were presented was confusing. We have now changed the graphs to be more clear.

5. In Fig 4A, please double check the indicated genotypes. Is the animal on the left L/L or wt/wt for Zbp1?

The animal on the left is Zbp1^{L/L} but not RIPK1^{E-KO}, presented just to show that exclusive expression of the long isoform does not cause skin pathology on its own. We realised that the way the genotypes were listed on this figure could confuse the reader to believe that all mice shown were RIPK1^{E-KO}, therefore, we have changed the labelling to make it more clear and avoid misunderstanding

6. Line 294: insert 'and' between caspase-3 and caspase-8

Will thank the reviewer for pointing out this typo, which we have now corrected in the revised manuscript.

Referee #2

In the study, the authors generated mouse models expressing exclusively either ZBP1-L (full length) or ZBP1-S (short isoform) endogenously. Using these mice and cells, the authors show that ZBP1-S regulates ZBP1-L function in vitro and in vivo. ZBP1-S restricts ZBP1-L-mediated cell death and inflammation-associated pathologies. The study and the mechanisms of ZBP1-S in regulating ZBP1-L are interesting. I have a few critical comments. Please see below.

1. The expression of the ZBP1-S in mice has been reported/discussed in previous studies (isoform-2 of the ZBP1), and authors have ignored this or were not aware of the existing literature (PMID: 32350114 <https://pubmed.ncbi.nlm.nih.gov/32350114/>; PMID: 32296175 <https://pubmed.ncbi.nlm.nih.gov/32296175/>, one of their own studies).

It has been demonstrated that the deletion of the Za2 domain of ZBP1 abolishes ZBP1-S form expression. (PMID: 32350114 <https://pubmed.ncbi.nlm.nih.gov/32350114/>).

Will thank the reviewer for their comprehensive assessment of our manuscript and their insightful comments. We did not intend to give the impression that the expression of the two isoforms was not observed earlier, as we and others have published experiments clearly demonstrating the expression of the long and short isoforms in different cell types. We have now included this information in the introduction and adjusted the text to make this point clear to the reader.

2. Whether human cells express the ZBP1-S isoform has not been explored. This needs to be tested for biological relevance. Most of the human cell lines may not express ZBP1, and thus, the authors should consider isolating PBMCs (or human monocyte/macrophage cells) to test the ZBP1-L and ZBP1-S expression.

As discussed above to a similar comment of reviewer 1, the role of ZBP1 in humans is a topic of intense investigation, but remains poorly understood. The human ZBP1 gene exhibits a complex alternative splicing pattern, with at least 7 predicted isoforms annotated in Uniprot (Q9H171). One of these isoforms (Q9H171-5) contains only the second Z α domain and lacks RHIMs, thus resembling the mouse ZBP1-S. However, it remains unclear whether all of these isoforms are expressed at the protein level in human cells and tissues and whether they perform different functions. This is largely due to the fact that it is very difficult to detect endogenous ZBP1 protein expression in most human cell types. We have included a short text at the beginning of the discussion section of our manuscript outlining the complex splicing pattern of human ZBP1. As the current literature on human ZBP1 is scarce and confusing, we opted not to discuss studies on human ZBP1 as we felt this would take too much space and might be counterproductive as we believe that for several of these studies it will be important to be reproduced across different labs before firm conclusions can be drawn. Overall, while we agree that studying the splicing pattern and functionally characterising the different isoforms in human cells will be very interesting and important, we respectfully suggest that this is outside the scope of the current manuscript which focuses on the functional characterisation of the two main isoforms in mice.

3. The mechanism shows that ZBP1-S antagonises the activation of ZBP1-L by competing for binding to Z-RNA. The only evidence provided was J2-antibody pull-downs. However, dsRNA pull down by J2-antibody does not provide direct evidence of ZBP1-S antagonizing ZBP1-L by restricting Z-RNA binding. The authors should perform pull-down experiments with Z-nucleic acid (Z-NA) specific antibody as used in multiple studies (Mouse monoclonal anti-Z-NA antibody, clone Z22) and probe for detection of ZBP1-S and ZBP1-L.

The reviewer correctly points out that the J2 antibody binds dsRNA and is not specific for Z-RNA. We are using this antibody because we showed in our previous studies that ZBP1 co-immunoprecipitates with dsRNA in a manner that depends on the presence of functional Z α domains. Our interpretation of these results is that some dsRNA species contain stretches of Z-RNA, which is what ZBP1 binds to. We are of course aware of the existence of the Z22 antibody raised against Z-DNA, which some groups have reported detects and immunoprecipitates Z-RNA. We have extensively tried to use the Z22 antibody to detect Z-RNA in cells both by immunofluorescence and immunoprecipitation, however, we were unable to obtain specific binding. At this point, we cannot explain why this antibody that has been reported to work in other labs does not work in our hands. However, unfortunately, because of this we cannot provide any data on ZBP1 isoform interaction with Z-NA.

4. The authors did not cite the literature appropriately. In page-2, lines 77 & 81, the authors should cite the following reference (PMID: 32350114, <https://pubmed.ncbi.nlm.nih.gov/32350114>). This study generated mice specifically lacking Za2 domain and showed Z-RNA binding specific regulation of ZBP1 functions.

Indeed, we missed to cite this study that also showed that Z α 2 deletion prevented perinatal lethality in *Ripk1^{mR/mR}* mice. We have now cited this reference in the revised manuscript.

5. Page no-3, Line-126: Authors stated, "whereas we did not detect expression of either isoform in unstimulated BMDM." Authors should be cautious while writing such sentences. Previous studies show the basal expression of ZBP1 in unstimulated BMDMs. This sentence needs to be corrected.

We are puzzled by this comment. Our statement simply reports the results of our experiments, namely that we did not detect ZBP1 expression in unstimulated BMDMs (Fig. 1D). This has also been shown by other studies, as for example the manuscript by Kesavardhana et al mentioned by the reviewer above which shows in Fig. 1B absence of ZBP1 expression in unstimulated BMDMs. Of course we are aware

of other reports showing ZBP1 expression in unstimulated BMDMs, and we believe these differences are likely due to the culture conditions that in some instances can induce ZBP1 expression. However, when we refer to our experiments, we do not understand the comment that we should be cautious when we simply describe our results and we see no reason to change this statement.

Referee #3

General summary and opinion about the principal significance of the study, its questions and findings:

Nagata, Shafer, Wachsmuth & Pasparakis use mouse genetics and cellular models to demonstrate that the stoichiometry between two isoforms of mouse ZBP1 regulates downstream apoptosis and necroptosis. The short mouse isoform, ZBP1-S, does not contain the two/three RHIMs that are present in the long mouse isoform and that are required to mediate signaling. The authors propose a competition model whereby ZBP1-S sequesters Z-nucleic acids without forming a complex with ZBP1-L. In general the study is solid; especially the mouse genetic studies are very convincing and well-controlled; i.e. data from ZBP1 knockout cells and heterozygous crosses contribute to the quality of the study. Importantly, this work also confirms an independent study now published in Cell Reports (PMID 38748877). I recommend to address the following concerns to solidify some conclusions related to the mechanism by which ZBP1-S inhibits ZBP1-L and to validate (or not) data from PMID 38748877.

Specific major concerns essential to be addressed to support the conclusions:

- Is ZBP1-S unique to mouse cells? If so, it is important to stress this point. I recommend to add this to the title, make it clearer in the abstract (i.e. on line 29-30 in the final sentence of the abstract) and throughout the main text. I find this an important point since it was not openly addressed in PMID 38748877.

As discussed above to a similar comment of reviewers 1 and 2, the role of ZBP1 in humans is a topic of intense investigation, but remains poorly understood. The human ZBP1 gene exhibits a complex alternative splicing pattern, with at least 7 predicted isoforms annotated in Uniprot (Q9H171). One of these isoforms (Q9H171-5) contains only the second Z α domain and lacks RHIMs, thus resembling the mouse ZBP1-S. However, it remains unclear whether all of these isoforms are expressed at the protein level in human cells and tissues and whether they perform different functions. This is largely due to the fact that it is very difficult to detect endogenous ZBP1 protein expression in most human cell types. We have included a short text at the beginning of the discussion section of our manuscript outlining the complex splicing pattern of human ZBP1. As the current literature on human ZBP1 is scarce and confusing, we opted not to discuss studies on human ZBP1 as we felt this would take too much space and might be counterproductive as we believe that for several of these studies it will be important to be reproduced across different labs before firm conclusions can be drawn. Overall, while we agree that studying the splicing pattern and functionally characterising the different isoforms in human cells will be very interesting and important, we respectfully suggest that this is outside the scope of the current manuscript which focuses on the functional characterisation of the two main isoforms in mice.

- Does ZBP1-S have a shorter half-life compared to ZBP1-L as proposed in PMID 38748877? It would be important to test (e.g. using a simple CHX chase assay). Even if the results do not match, this should be reported.

We performed CHX chase and proteasome inhibitor treatment experiments and include the results in the revised manuscript (Fig EV 1C-D). These experiments provided some support that ZBP1-S has a shorter half-life compared to ZBP1-L, although in our assays the difference between the two isoforms was much less impressive compared to the studies reported by Cai et al.

- Human ZBP1 also induces NF- κ B activation, at least in an assay whereby the protein is inducibly expressed using a Tet-On system (PMID 36268590 <https://pubmed.ncbi.nlm.nih.gov/36268590/>). The authors should refer to this work in the introduction (e.g. on lines 50-51). Does mouse ZBP1 similarly induce NF- κ B and does ZBP1-S regulate this process? The authors have developed the perfect system to test this idea. For example, RT-qPCR and/or ELISA for NF- κ B activated genes after addition of doxycycline would do the job.

We are familiar with the study by Peng et al showing that doxycycline-inducible expression of human ZBP1 induced NF- κ B-dependent gene expression. We have also assessed whether doxycycline-induced expression of mouse ZBP1-L can induce NF- κ B-dependent gene expression in a previous study, and found that this is not the case. These results are reported in a preprint publication (Koerner *et al*, 2024), which we now cite in the revised manuscript and briefly discuss that we have been unable to detect NF- κ B activation by ZBP1-L therefore we could not assess the role of the short isoform in this process.

- Figure 2D and related Western blot: why is there no detectable cell death in wild type cells after IFN- α /em+et stimulation? Is it a titration issue or the fact that IFN- γ (as described by the authors in PMID 32296175), but not IFN- α triggers ZBP1-dependent cell death in these cells? The authors should address this experimentally.

The reviewer correctly points out that in this particular experiment we did not detect cell death in wild type LFs stimulated with IFN α +Em+Et. This is mainly a titration issue as the main aim of this experiment is to reveal the difference between cells expressing only ZBP1-L and cells expressing both the long and the short isoform. Under these conditions, we did see some cell death in wild type cells in some of the experiments, which was always much less compared to the *Zbp1^{L/L}* cells. We have now replaced the CDA shown in Fig 2D with another replicate that shows some cell death in wild type cells to avoid giving the impression that wild type cells do not die at all under these conditions.

- Readers that are less familiar with ZBP1 signaling and how apoptosis vs necroptosis is regulated may be confused about the hierarchy of ZBP1 signaling and the choice for IFN- α /Em/Et (necroptosis) and KPT-330 (apoptosis + necroptosis) to activate ZBP1. As far as I know, it is not clear how the nuclear export inhibitor KPT-330 activates ZBP1. It would be more logical to test IFN- α /smac mimetic (e.g. BV6)/Et to try and induce ZBP1-mediated apoptosis in analogy with TNF signaling.

We are using these two experimental systems, namely IFN α +Em+Et and IFN α +KPT, as we have established these assays in our lab and have demonstrated that these stimulations induce ZBP1-mediated cell death that depends on functional Z α -domains (Jiao *et al*, 2020). Indeed, the mechanism by which nuclear export inhibitors activate ZBP1-mediated apoptosis and necroptosis remains elusive, but this does not affect the conclusions drawn from our experiments. The aim of these experiments is to demonstrate that expression of the short isoform inhibits ZBP1-L-mediated cell death, which is clearly shown in the provided data. We have not extensively tried using smac mimetics to sensitize cells to ZBP1-mediated cell death and we are not confident that these will induce only ZBP1-mediated apoptosis or necroptosis.

- Figure 6 is the weakest part of the study. Several controls are missing and should be included. Did the co-IP in figure 6C work? For instance, does RIPK3 co-IP under conditions of em or KPT-330? The reverse co-IP (i.e. ZBP1-S IP) should also be performed in these conditions. Can the authors include a control to show that the J2 IP worked (e.g. dot blotting for dsRNA). The authors state that ZBP1-S competes with ZBP1-L to bind to Z-RNA. Have the authors tried to IP Z-RNA/DNA using Z22? Moreover, from the figure legend, it appears that the experiment in figure 6D has only been performed once. This experiment should be repeated.

The reviewer raises different issues here. First, the IP in figure 6C was performed to assess whether the long and short isoforms interact. The results clearly showed that the short isoform was not detected in the immunoprecipitates of the long isoform, which demonstrates that the two isoforms do not form stable complexes in cells. The co-IP worked, as shown by the fact that we could detect RIPK3 co-immunoprecipitating with ZBP1-L in these assays. The RIPK3 blots are now included in the figure in the revised manuscript. We also repeated the co-IP using anti-mCherry antibodies to immunoprecipitate the short isoform and obtained similar results as we could not detect the long isoform in the ZBP1-S IP. These results are now included in the revised manuscript.

Regarding the J2 IP, this immunoprecipitation clearly worked as we could detect ZBP1-L and ZBP1-S co-immunoprecipitating with dsRNA. As discussed also above in our response to a comment from reviewer 1, our J2 antibody pulldown experiments showed that both the long and short isoforms of ZBP1 bind to dsRNA via their Z α domains. We have now repeated this experiment several times and our overall conclusion is that these assays do not provide strong evidence that ZBP1-S expression reduces the binding of ZBP1-L to dsRNA. While we still favour the hypothesis that ZBP1-S suppresses

ZBP1-L-induced cell death by competing for binding to Z-NAs, this experiment cannot provide a clear answer to this question. We have adjusted the text in the discussion to draw the readers' attention to this point. Lastly, the reviewer asks whether we have tried to IP Z-NAs using the Z22 antibody. As discussed in our responses to a similar comment from reviewer 2 above, the J2 antibody binds dsRNA and is not specific for Z-RNA. We are using this antibody because we showed in our previous studies that ZBP1 co-immunoprecipitates with dsRNA in a manner that depends on the presence of functional Z α domains. Our interpretation of these results is that some dsRNA species contain stretches of Z-RNA, which is what ZBP1 binds to. We are of course aware of the presence of the Z22 antibody raised against Z-DNA, which some groups have reported detects and immunoprecipitates Z-RNA. We have extensively tried to use the Z22 antibody to detect Z-NA in cells both by immunofluorescence and immunoprecipitation, however, we were unable to obtain specific binding. At this point, we cannot explain why this antibody that has been reported to work in other labs does not work in our hands. However, unfortunately, because of this we cannot provide any data on ZBP1 isoform interaction with Z-NA.

Minor concerns that should be addressed:

- Line 92-93: Karki et al 2021 (PMID 34686350) also report that ADAR1 inhibits ZBP1. It would be fair to also refer to this study.

This sentence specifically refers to the role of ZBP1 in mediating perinatal lethality in mice hemizygotously expressing ADAR1 with mutation of its Z α domain, which has been demonstrated by the three cited papers. The study by Karki et al reports on the role of ZBP1 in a different system, mediating anti-cancer immunity induced by ADAR1 ablation in myeloid cells, this is the reason for not citing it. With the ZBP1 literature exponentially growing, it is not possible to cite all studies reporting different functions, which is more the task of a comprehensive review article and not a research paper like ours focusing on a specific aspect of ZBP1 biology.

- Line 125/Figure 1C/D and line 322/Figure 6D: Quantitative statements based on Western blots should be avoided. For example transfer times affect the ratios of larger vs smaller proteins on the blotting membrane.

We fully agree that quantitative statements based on western blots should be avoided as these assays do not accurately report expression levels. In our manuscript we refrained from making quantitative statements in describing the results. For example, in lines 125-130 we wrote: "Primary LFs from wild type mice showed **weak basal expression** of ZBP1-L, whereas **we did not detect expression** of either isoform in unstimulated BMDMs (Fig. 1C, D). However, after stimulation with IFN α for 24h both LFs and BMDMs **showed robust expression** of ZBP1-L and ZBP1-S (Fig. 1C, D), showing that both isoforms are induced by IFNs. Furthermore, both the long and short ZBP1 isoforms **were expressed** in the thymus and spleen from wild type mice (Fig. 1E)." These sentences do not make quantitative statements but describe the results of the western blots in a neutral and objective manner. We could not find any quantitative statements in line 322 when we refer to figure 6D.

- Line 195, figure 3B: Zbp1S/S rescued mice are not on the graph

The reason we had not included RIPK1^{E-KO} Zbp1^{S/S} in this graph is that we had only young mice at the time of submission of the original manuscript. We have now observed 6 mice up to the age of 150 days and none of these animals developed skin lesions, further supporting that Zbp1^{S/S} mice behave like Zbp1^{-/-} in this setting. These results are now included in the figure.

- Please provide gating strategies for figures 3F and 4F

The gating strategy for the flow cytometry experiment is presented in new Fig EV2.

- Line 305: why would ZBP1-S form heterodimers with ZBP1-L? Could also be hetero-oligomers. This is a good point. We agree and have edited this sentence accordingly.

- Line 365: the preprint has now been published (PMID 38748877) We have updated the citation.

• Could the authors change the color schemes and/or change symbols in the cell death graphs? The different conditions are hard to discern.

We have tried to use colors that are also appropriate for color blind people which reduces the choice, but have made an effort to improve the clarity of the graphs.

• Figures 3B and 4B, the Y-axis should depict "dead (%)" instead of "survival (%)".

We agree that these graphs presented the results in a confusing way. We have changed the graphs to more clearly depict the findings.

• Line 305-306: Confocal fluorescent microscopy can determine the localization of ZBP1-S and ZBP1-L. We have considered using fluorescent microscopy to assess the localization of the two isoforms. We have performed some preliminary assays using the fluorescently tagged proteins and detected difused expression throughout the cells and decided not to follow this further. Considering that a comprehensive analysis and quantification of subcellular localisation using advanced microscopy techniques will require an extensive period of time, we respectfully suggest that this is not an essential experiment to include in the current manuscript.

Any additional non-essential suggestions for improving the study (which will be at the author's/editor's discretion):

I wonder whether a more condensed form of the manuscript would be better. Figure 6 is thus far less substantiated compared to figure 5. Since both figures address the mechanism by which ZBP1-S suppresses ZBP1-L function both figures could be merged.

We appreciate the suggestion of the reviewer. However, we are worried that trying to fit all the data in Figure 5 will result in a very busy figure with small panels that might be difficult for the readers to follow. We would prefer to keep the data in Figure 6, but we'd of course be prepared to consolidate these to one figure if needed.

Referee #4

In this study, Nagata et al. investigated the roles of the two ZBP1 isoforms, the short isoform (ZBP1-S) containing just the Za domains, and the long isoform (ZBP1-L) containing the Za domains and C-terminal RHIM domains. They show that ZBP1-S acts as an endogenous inhibitor of ZBP1-L. Using newly generated mouse models, the authors showed that loss of ZBP1-S exacerbated skin inflammation induced by ZBP1-mediated cell death in RIPK1E-KO mice. The study also used cellular systems to suggest that ZBP1-S inhibits ZBP1-L-induced cell death by competing for binding to Z-nucleic acids. Overall, these findings provide insights into the autoregulation of ZBP1. While another paper reporting similar findings regarding these two isoforms was recently published in Cell Reports (PMID: 38748877), the current study includes in vivo analyses, which extend beyond the Cell Reports study. The authors should perform additional analyses and consider alternative interpretations of their data to strengthen their study.

Major Comments

1. The authors observed higher expression of ZBP1-L when ZBP1-S was removed (Fig. 2C). This effect complicates the interpretation of cell death phenotypes. The authors should consider an alternative interpretation of their data: that the higher cell death observed might be due to increased ZBP1-L levels, with ZBP1-S controlling ZBP1-L expression. The authors should perform additional control experiments to test this possibility, such as designing transcript-specific primers or probes to test relative transcript levels of ZBP1-L and ZBP1-S.

Indeed, our experiments suggest that the expression of each isoform is increased when the other isoform cannot be generated, namely *Zbp1^{S/S}* cells express more ZBP1-L and *Zbp1^{L/L}* cells express more ZBP1-S compared to wild type cells. However, this rather small increase in ZBP1-L expression could not explain the dramatic enhancement of ZBP1-L-induced cell death observed in the absence of the short isoform. Our experiments in dox-inducible cellular systems also provide strong support that the increased death observed in cells expressing exclusively ZBP1-L is not due to elevated ZBP1-L expression levels but rather caused by the absence of the short. For example, in Fig. EV2D the amount

of cell death induced by similar levels of ZBP1-L is enhanced in the absence and suppressed in the presence of ZBP1-S but not ZBP1-S with mutated $Z\alpha$ domains. Since in this setting the two isoforms are expressed by different plasmid-based vectors, the effect of ZBP1-S expression cannot be mediated by regulation of the mRNA levels of ZBP1-L. We have measured the mRNA expression of the long and short isoforms of ZBP1 in cells also in response to IFN stimulation. These results are presented in Fig. EV1B and show that both isoforms are robustly induced by IFN. This is different from the study by Cai et al that reported that IFN stimulation induces much higher expression of the short isoform compared to the long isoform.

2. In addition to its role in cell death, ZBP1 has cell death-independent, pro-inflammatory functions via RIPK1/3 (PMID: 36268590 <https://pubmed.ncbi.nlm.nih.gov/36268590/>). The authors should determine the function of ZBP1-S in this context, including its effects on inflammatory gene expression in both fibroblast and myeloid cells.

As discussed in response to a similar comment from reviewer 3 above, we are familiar with the study by Peng et al showing that doxycycline-inducible expression of human ZBP1 induced NF- κ B-dependent gene expression. We have also assessed whether doxycycline-induced expression of mouse ZBP1-L can induce NF- κ B-dependent gene expression in a previous study, and found that this is not the case. These results are reported in a recent publication (Koerner *et al.*, 2024), which we now cite in the revised manuscript and briefly discuss that we have been unable to detect NF- κ B activation by ZBP1-L therefore we could not assess the role of the short isoform in this process.

3. To draw conclusions about the effect of ZBP1-S on inflammation specifically, the authors should test inflammatory cytokine release by ELISA from both the in vitro and in vivo studies.

As explained in response to the previous comment of the reviewer, in our experimental systems we were unable to detect activation of NF- κ B-dependent inflammatory gene expression by expression of ZBP1 (Koerner *et al.*, 2024). Therefore, measuring cytokine release by ELISA in these cells would not provide meaningful data. Regarding the in vivo studies, in our experience the expression of cytokines and chemokines in the skin of RIPK1^{E-KO} mice correlates with the severity of the lesions, which are triggered by necroptosis of keratinocytes as shown by the fact that the mice are fully rescued by MLKL deficiency. Since our immunostainings and FACS assays clearly demonstrate the effects of ZBP1-L and ZBP1-S in skin inflammation in RIPK1^{E-KO} mice, measuring cytokine expression by ELISA will not provide essential information.

Minor Comments

1. While the authors conclude that ZBP1-S may outcompete ZBP1-L for binding to nucleic acids, the data to support this conclusion in Figure 6D are not strong, and the mechanism through which this would happen is not clear. Is ZBP1-L completely dispensable for the interaction of ZBP1-S with nucleic acids? What are the consequences of ZBP1-L being partially pulled down along with the J2 antibody? To make this conclusion, the authors should assess the KD for each of the isoforms to interact with nucleic acids.

We agree with the reviewer that while the J2 antibody IP results indicate that the short isoform may bind Z-RNA better than the long, the data is not strong enough to draw a firm conclusion. As discussed in the manuscript, we favour the model that ZBP1-S acts by competing with the long isoform for access to nucleic acids. This is based primarily on the findings that ZBP1-S requires functional $Z\alpha$ domains in order to be able to inhibit ZBP1-L-induced cell death. Assessment of the affinity and strength of interaction of the two isoforms with Z-RNA would indeed be an elegant approach to provide more information on the mechanism. However, there are a number of technical issues that make this experiment unfeasible at present. These relate to the fact that full-length ZBP1 is very difficult to produce as a recombinant protein as it is not soluble, most likely because of the highly unstructured C-terminal fragment containing the RHIMs. Another issue relates to the uncertainty as to the nature of the physiological endogenous ligands that bind and activate ZBP1. Nevertheless, we believe that our data fully support our conclusion that ZBP1-S acts as an inhibitor of the long isoform to fine-tune ZBP1-mediated responses, much like the E3 protein of poxviruses, which employs its $Z\alpha$ domain to suppress ZBP1-mediated cell death in response to virus infection (Koehler *et al.*, 2021).

2. The authors should discuss the possible functions of ZBP1-S and ZBP1-L in different cancer models.

For example, what could be the phenotype of these isoforms in tumor surveillance? Additionally, the authors should discuss the relevance of these findings for humans.

We understand that the possible role of ZBP1 in cancer is a topic that has gained increased attention recently. However, with due respect, considering the restricted space and the focus of our manuscript, we suggest that discussing the potential role of the two isoforms in different cancer models and tumor surveillance is beyond the scope of the current manuscript. As discussed above to a similar comment of the other reviewers, the role of ZBP1 in humans is a topic of intense investigation, but remains poorly understood. The human ZBP1 gene exhibits a complex alternative splicing pattern, with at least 7 predicted isoforms annotated in Uniprot (Q9H171). One of these isoforms (Q9H171-5) contains only the second Z α domain and lacks RHIMs, thus resembling the mouse ZBP1-S. However, it remains unclear whether all of these isoforms are expressed at the protein level in human cells and tissues and whether they perform different functions. This is largely due to the fact that it is very difficult to detect endogenous ZBP1 protein expression in most human cell types. We have included a short text at the beginning of the discussion section of our manuscript outlining the complex splicing pattern of human ZBP1. As the current literature on human ZBP1 is scarce and confusing, we opted not to discuss studies on human ZBP1 as we felt this would take too much space and might be counterproductive as we believe that for several of these studies it will be important to be reproduced across different labs before firm conclusions can be drawn. Overall, while we agree that studying the splicing pattern and functionally characterising the different isoforms in human cells will be very interesting and important, we respectfully suggest that this is outside the scope of the current manuscript which focuses on the functional characterisation of the two main isoforms in mice.

3. There is a regulatory relationship between ADAR1 and ZBP1. Assessing the correlation between ADAR1 expression and ZBP1-S may provide a broader view of ZBP1 regulation.

We agree that the relationship between ZBP1 and ADAR1 is a very interesting topic and that exploring the role of ZBP1-S in this context will be important. We are indeed doing experiments in this direction but these involve complex genetic crosses that will take a long time and therefore cannot be included in the current manuscript.

4. In the introduction, the authors note that ZBP1 mediates heatstroke-induced cell death; however, this is currently controversial in the field (PMID: 35511979 <https://pubmed.ncbi.nlm.nih.gov/35511979/>, 38409108 <https://pubmed.ncbi.nlm.nih.gov/38409108/>).

We are aware that the recent paper by the Kanneganti group reported that ZBP1 was not required for heatstroke induced cell death in BMDMs. However, we have own data that in fibroblasts ZBP1 is indeed required for heatstroke-induced cell death, supporting the conclusions of the first study. We prefer to keep the citation as it is.

5. The survival graphs (Figure 3B and 4B) are difficult to follow, as the y-axis indicates it is % survival, but the actual data plotted seem to be % with disease or % dead.

We agree that the graphs were presented in a confusing manner. This has been corrected in the revised manuscript.

6. Additional controls are needed in some of the western blots. For example, in Fig. 6D, a control for ZBP1mZ α -S is missing to show that ZBP1-S can bind dsRNA. Additionally, in Fig 2E and 2G, the total MLKL band is highly variable. The authors should explain this.

We have now included ZBP1mZ α -S in the experiment in Fig. 6D. Regarding MLKL, its expression is induced by IFNs and this is the reason for the variable expression levels in Figs 2E and G.

References

Jiao H, Wachsmuth L, Kumari S, Schwarzer R, Lin J, Eren RO, Fisher A, Lane R, Young GR, Kassiotis G *et al* (2020) Z-nucleic-acid sensing triggers ZBP1-dependent necroptosis and inflammation. *Nature* 580: 391-395

Koehler H, Cotsmire S, Zhang T, Balachandran S, Upton JW, Langland J, Kalman D, Jacobs BL, Mocarski ES (2021) Vaccinia virus E3 prevents sensing of Z-RNA to block ZBP1-dependent necroptosis. *Cell Host Microbe* 29: 1266-1276 e1265

Koerner L, Wachsmuth L, Kumari S, Schwarzer R, Wagner T, Jiao H, Pasparakis M (2024) ZBP1 causes inflammation by inducing RIPK3-mediated necroptosis and RIPK1 kinase activity-independent apoptosis. *Cell Death Differ*

Dear Manoli,

Thank you for the submission of your revised manuscript to The EMBO Journal and your patience during its peer review. It has now been seen by three of the original referees who previously assessed the earlier version of your manuscript, and we have received their comments (included below). I am glad to say that all referees have rated the novelty and significance of the work highly, and they all recognize that the majority of their previously raised concerns have been satisfactorily addressed. Referee #1 has no further comments, while referees #2 and #3 list a number of suggestions for further improvement of the text (some of which we have already discussed before the submission of your revision), which I would kindly ask you to take into consideration while preparing a final revised version of your manuscript. Please also include in your resubmission a point-by-point response to these remaining comments describing any changes to the manuscript.

From the editorial side, there are also a few changes and corrections that we need from you before we can proceed with acceptance of the manuscript for publication in The EMBO Journal:

- Please note that the full funding information should be entered in our manuscript handling system during resubmission of your manuscript, and it should be identical to the information provided in the Acknowledgements section of your manuscript. The information related to funding from the Alexander von Humboldt foundation is currently missing in our online system.
- Please change the heading "Methods" to "Materials and Methods".
- Please change the heading of your conflict-of-interest statement to "Disclosure and competing interests statement".
- The author contributions statement should be removed from the manuscript file. Instead, we use CRediT to specify the contributions of each author in the journal submission system. Please feel free to use the free text box to provide more detailed descriptions during submission. See also our guide to authors for more information:
<https://www.embopress.org/page/journal/14602075/authorguide#authorshipguidelines>.
- We noticed that there is a callout for Table EV1, which has not been uploaded. Could you please check again and correct the callout or upload the missing Table?
- Please provide in the Materials and Methods (in the paragraph describing the mouse experiments) the reference number for approval of these experiments.
- Please zip together all Source Data for Figures EV1 and EV3 in a single folder named "EV Figures Source Data".
- Please complete and upload the Source Data checklist that our Source Data coordinator (Hannah Sonntag) has sent you.
- Please note that EMBO press papers are accompanied online by:
 - A) a short (2 sentences) summary of the findings and their significance,
 - B) 2-5 short bullet points highlighting the key results, and
 - C) a synopsis image in .jpg or .png format that is exactly 550 pixels wide and 300-600 pixels high (the height is variable). Please note that the text needs to be legible at the final size. Please upload this information along with your revised manuscript (the text for A and B should be provided in a separate Word file).
- Please note that the legends for Figures 2e-f are not provided in the sequential manner (i.e. the legend for Figure 2f is provided before the legend of Figure 2e). This needs to be rectified.
- Please note that the legends for Figures 5d-e are not provided in the sequential manner (i.e. the legend for Figure 5e is provided before the legend of Figure 5d). This needs to be rectified.
- Please note that the legends for Figures EV 3d-e are not provided in the sequential manner (i.e. the legend for Figure EV 3e is provided before the legend of Figure EV 3d). This needs to be rectified.
- Please note that the exact p values should be provided in the legends of Figures 3e-f; 4c, e-f.
- Please note that information related to "n" is missing in the legends of Figures 3e-f; 4c, e-f.
- Please note that the error bars are not defined in the legends of Figures 3e-f; 4c, e-f; 5c, e; EV 1b.
- In a standard image analysis, we detected possible aberrations in your Figure set that we would like to discuss with you:
 1. Re-use of mouse pictures was detected between Figure 3A and Figure 4A (control RIPK1 E-KO mice). This should be corrected or, if re-use is experimentally justified, it should be explicitly stated in the Figure legends.

2. Similarities were detected between cells in Figure 3D control - HE vs. Trichrome (subimages match). Could you please check again and clarify whether these are the same or different cells?
3. Similarities were detected between cells in Figure 4D RIPK1 E-KO Zbp1 WT/L H&E vs. Trichrome. Could you please check again and clarify whether these are the same or different cells?
4. Similarities were detected in the blot shown in Fig. EV1C (WT_1 vs. WT_3). Can you please confirm that these are different samples?

Please also note that as part of the EMBO publications' Transparent Editorial Process, The EMBO Journal publishes online a Peer Review File along with each accepted manuscript. This File will be published in conjunction with your paper and will include the referee reports, your point-by-point response and all pertinent correspondence relating to the manuscript. You can opt out of this by letting the editorial office know (contact@embojournal.org). If you do opt out, the Peer Review File link will point to the following statement: "No Peer Review File is available with this article, as the authors have chosen not to make the review process public in this case."

We look forward to seeing a final version of your manuscript as soon as possible. Please use this link to submit your revision: <https://emboj.msubmit.net/cgi-bin/main.plex>

Best regards,

Ioannis

Referee #1:

The authors addressed my comments fully. I congratulate the authors on this nice piece of work and recommend swift publication.

Referee #2:

The revised version of the manuscript addresses my comments. However, the authors still need to address the following comment which is critical for the ZBP1 and Z-RNA field.

The authors mentioned that- "We have extensively tried to use the Z22 antibody to detect Z-RNA in cells both by immunofluorescence and immunoprecipitation, however, we were unable to obtain specific binding. At this point, we cannot explain why this antibody that has been reported to work in other labs does not work in our hands. However, unfortunately, because of this we cannot provide any data on ZBP1 isoform interaction with Z-NA".

In this case, the authors should provide the data in the manuscript showing that Z22 antibody do not show specific binding. This is indeed critical for justifying why these experiments were performed with J2-antibody and let the cell death community aware of the reliability of Z22 antibody for future Z-RNA studies.

Referee #3:

The authors have satisfactorily addressed almost all of my concerns. I wish to congratulate the authors on this well-executed study. I listed some additional remarks below, which may be addressed in the final manuscript:

- I am not sure if Q9H171-5 maps to a corresponding transcript. Human Uniprot isoform Q9H171-6 (Ensembl transcript ID ENST00000541799.1) contains both Za1 and Za2, but not the RHIMs, better resembling mouse ZBP1S. I suggest to add this possible human isoform to the discussion.

- From Fig. 6C of the revised manuscript it seems that endogenous RIPK3 associates constitutively with overexpressed ZBP1-L-NG even in the absence of KPT-330 or Emricasan, although perhaps at lower levels compared to the stimulated conditions. It may be worth commenting on this. The reverse IP nicely provides further evidence that ZBP1-L and ZBP1-S do not enter the same complex at least not under the conditions that were used for the IP.
- IPs with Z22 are indeed technically challenging. The Zalpha domain mutant controls -both for the L and S isoforms are important controls for fig 6D of the revised manuscript. Based on the inclusion of these controls, I believe it is OK to infer that the dsRNA IP has worked even without a dot blot showing enrichment of dsRNA after J2 IP.
- The quantitative statement re. Western blots on line 322 of the original manuscript this reviewer was referring to was 'ZBP1-S may bind stronger to dsRNA'. This is resolved in the current version of the manuscript. Nevertheless I am still missing the rationale/evidence behind the statement that ZBP1-S may bind better to dsRNA, (lines 339 and 362 of the revised manuscript).
- Line 376: the fact mouse ZBP1 does not activate NF-kB may be related to a species difference. Indeed, the studies mentioned here showing ZBP1-dependent NF-kB activation were performed by overexpression of human ZBP1 in human cell lines.

Point by point response to the reviewer comments**Referee #1:**

The authors addressed my comments fully. I congratulate the authors on this nice piece of work and recommend swift publication.

We thank the reviewer for their insightful comments that helped improve our manuscript.

Referee #2:

The revised version of the manuscript addresses my comments. However, the authors still need to address the following comment which is critical for the ZBP1 and Z-RNA field.

The authors mentioned that- "We have extensively tried to use the Z22 antibody to detect Z-RNA in cells both by immunofluorescence and immunoprecipitation, however, we were unable to obtain specific binding. At this point, we cannot explain why this antibody that has been reported to work in other labs does not work in our hands. However, unfortunately, because of this we cannot provide any data on ZBP1 isoform interaction with Z-NA".

In this case, the authors should provide the data in the manuscript showing that Z22 antibody do not show specific binding. This is indeed critical for justifying why these experiments were performed with J2-antibody and let the cell death community aware of the reliability of Z22 antibody for future Z-RNA studies.

We thank the reviewer for their insightful comments.

We acknowledge the importance of publishing negative data and therefore understand the rationale behind the suggestion to include in the manuscript the results from our unsuccessful efforts to IP Z-RNA using the Z22 antibody. However, as we are sure the reviewer also knows, it is very hard to provide robust datasets and evidence documenting that an experiment did not work. For this reason, we do not have figure quality data to document the irreproducibility of the Z22 IP experiments to provide. It would take a major effort and a considerable amount of time to prepare and put together all the data that would be needed to demonstrate the lack of specificity, therefore we regret that we cannot include these negative results in the manuscript. However, we are continuing our efforts to establish the experimental conditions to effectively immunoprecipitate and study Z-RNA from cells and we do plan to publish our experience with the different antibodies in a dedicated manuscript in the future.

As the J2-antibody has extensively been used to detect dsRNA in the context of Z-nucleic acid research, we believe the J2-IP provides sufficient proof of dsRNA-ZBP1 interactions.

Referee #3:

The authors have satisfactorily addressed almost all of my concerns. I wish to congratulate the authors on this well-executed study. I listed some additional remarks below, which may be addressed in the final manuscript:

- I am not sure if Q9H171-5 maps to a corresponding transcript. Human Uniprot isoform Q9H171-6 (Ensembl transcript ID ENST00000541799.1) contains both Za1 and Za2, but not the

RHIMs, better resembling mouse ZBP1S. I suggest to add this possible human isoform to the discussion.

We thank the reviewer for their meticulous observations. To accurately address their comments, we have revised the coding hZBP1 isoforms currently supported by Uniprot.

While the human Uniprot isoform Q9H171-6 (248 aa long) contains both Za1 and Za2 domains, it is also, according to the Pfam database, predicted to contain the first RHIM (198-215 aa). This isoform resembles a recently reported artificial murine ZBP1 truncation construct (ZBP1ca) lacking RHIM2 and RHIM3, that has been shown to act as a constitutively active ZBP1 (Koerner et al., 2024). Therefore, the human Q9H171-6 isoform would in all likelihood have the opposite effect of the mouse ZBP1-S, enhancing cell death instead of mitigating it.

In contrast, the human Uniprot isoform Q9H171-5 (149 aa), contains only the first Za domain and lacks all RHIMs. Given that both human Z-DNA binding domains, Za1 and Za2, have been suggested to independently bind Z-DNA (Deigendesch et al., 2024), we postulate that this human ZBP1 isoform may behave similarly to mouse ZBP1-S and thus negatively regulate ZBP1-mediated signaling.

- From Fig. 6C of the revised manuscript it seems that endogenous RIPK3 associates constitutively with overexpressed ZBP1-L-NG even in the absence of KPT-330 or Emricasan, although perhaps at lower levels compared to the stimulated conditions. It may be worth commenting on this. The reverse IP nicely provides further evidence that ZBP1-L and ZBP1-S do not enter the same complex at least not under the conditions that were used for the IP.

We thank the reviewer for pointing out this observation. Indeed, we have observed also in our previous studies (Jiao et al., 2020; Koerner et al., 2024) that ZBP1 seems to associate with RIPK3 even in the absence of any treatment. The lack of any ensuing ZBP1-RIPK3-MLKL – induced necroptosis is suggested to be due to the negative regulation by RIPK1 and caspase-8.

- IPs with Z22 are indeed technically challenging. The Zalpha domain mutant controls -both for the L and S isoforms are important controls for fig 6D of the revised manuscript. Based on the inclusion of these controls, I believe it is OK to infer that the dsRNA IP has worked even without a dot blot showing enrichment of dsRNA after J2 IP.

We thank the reviewer for sharing their experience with Z22 IPs, it is helpful to know that other labs also struggle with these experiments.

- The quantitative statement re. Western blots on line 322 of the original manuscript this reviewer was referring to was 'ZBP1-S may bind stronger to dsRNA'. This is resolved in the current version of the manuscript. Nevertheless I am still missing the rationale/evidence behind the statement that ZBP1-S may bind better to dsRNA, (lines 339 and 362 of the revised manuscript).

We thank the reviewer for pointing this out, we have now re-phrased this sentence to avoid making a quantitative statement on the strength of the interaction of the ZBP1 isoforms with dsRNA. This sentence now writes : *“Although the detailed molecular mechanism by which ZBP1-S inhibits the activation of the long isoform remains to be fully elucidated, our results*

suggest that the short isoform likely acts as a decoy competing with ZBP1-L for binding to cognate ligands.”

- Line 376: the fact mouse ZBP1 does not activate NF- κ B may be related to a species difference. Indeed, the studies mentioned here showing ZBP1-dependent NF- κ B activation were performed by overexpression of human ZBP1 in human cell lines.

Indeed, the study by Peng et al used human ZBP1 overexpression in human cells, while our experiments have been performed in mouse cells using mouse ZBP1. Species specificity could be one reason for the fact we did not observe NF- κ B activation in our experiments. As this is a rather new field, replicating these studies in different labs using different experimental systems will be needed before a consensus emerges.

Dear Manoli,

Congratulations on an excellent manuscript, I am very pleased to inform you that it has been accepted for publication in The EMBO Journal. Thank you very much for your comprehensive responses to the referee concerns and for addressing all editorial requests.

Your manuscript will now be processed for publication by EMBO Press. It will be copy edited and you will receive page proofs prior to publication. Please note that you will be contacted by Springer Nature Author Services to complete licensing and payment information.

If you have any questions, please do not hesitate to contact the Editorial Office. Thank you for your contribution to The EMBO Journal. Working with you has been a pleasure!

Best wishes,

Ioannis
